# Intercomparison of MAX-DOAS Vertical Profile Retrieval Algorithms: Studies using Synthetic Data.

Udo Frieß[1], Steffen Beirle[2], Leonardo Alvarado Bonilla[3], Tim Bösch[3], Martina M. Friedrich[4], François Hendrick[4], Ankie Piters[5], Andreas Richter[3], Michel van Roozendael[4], Vladimir V. Rozanov[3], Elena Spinei[6,7], Jan-Lukas Tirpitz[1], Tim Vlemmix[5], Thomas Wagner[2], and Yang Wang[2]

[1]Institute of Environmental Physics, University of Heidelberg, Heidelberg, Germany
[2]Max Planck Institute for Chemistry, Mainz, Germany
[3]Institute of Environmental Physics, University of Bremen, Bremen, Germany
[4]Royal Belgian Institute for Space Aeronomy (BIRA-IASB), Brussels, Belgium
[5]Royal Netherlands Meteorological Institute (KNMI), De Bilt, The Netherlands
[6]NASA-Goddard, Greenbelt, Maryland, USA
[7]now at Virginia Tech, Blacksburg, Virginia, USA

*Correspondence to:* Udo Frieß (udo.friess@iup.uni-heidelberg.de)

**Abstract.** Multi-Axis Differential Optical Absorption Spectroscopy (MAX-DOAS) is a widely used measurement technique for the detection of a variety of atmospheric trace gases. Using inverse modelling, the observation of trace gas column densities along different lines of sight enables the retrieval of aerosol and trace gas vertical profiles in the atmospheric boundary layer using appropriate retrieval algorithms. In this study, the ability of eight profile retrieval algorithms to reconstruct vertical profiles is assessed on the basis of synthetic measurements. Five of the algorithms are based on the optimal estimation method, two on parametrised approaches, and one using an analytical approach without involving any radiative transfer modelling. The synthetic measurements consist of the median of simulated slant column densities of $O_4$ at 360 nm and 477 nm, as well as of HCHO at 343 nm and $NO_2$ at 477 nm, from seven datasets simulated by five different radiative transfer models. Simulations are performed for a combination of 10 trace gas and 11 aerosol profiles, as well as 11 elevation angles, 3 solar zenith and 3 relative azimuth angles. Overall, the results from the different algorithms show moderate to good performance for the retrieval of vertical profiles, surface concentrations and total columns. Except for some outliers, the root mean square difference between the true and retrieved state ranges between $(0.05-0.1)$ km$^{-1}$ for aerosol extinction, and $(2.5-5.0) \cdot 10^{10}$ molec/cm$^3$ for HCHO and $NO_2$ concentrations.

## 1 Introduction

The planetary boundary layer (PBL) is the part of the atmosphere that is in direct contact with the terrestrial biosphere. Its chemical composition is determined by anthropogenic and natural emissions. Monitoring of both chemical composition and aerosol content of the PBL is crucial for the understanding of the chemical and physical processes and the spatio-temporal evolution of PBL composition. A versatile tool for the monitoring of atmospheric trace gases and aerosol content of the PBL is the well-known Multi-Axis Differential Optical Absorption Spectroscopy (MAX-DOAS) (e.g. Hönninger et al., 2004; Wagner

et al., 2004; Heckel et al., 2005; Frieß et al., 2006; Platt and Stutz, 2008; Irie et al., 2008; Clémer et al., 2010; Wagner et al., 2011; Vlemmix et al., 2015b). It relies on the spectral analysis of scattered sunlight and enables the simultaneous detection of numerous trace gases, such as nitrogen dioxide ($NO_2$), formaldehyde (HCHO), nitrous acid (HONO), water vapor ($H_2O$), sulfur dioxide ($SO_2$), ozone ($O_3$), and halogen oxides. Measurements along different lines of sight, with elevation angles (EA) ranging from near the horizon to the zenith, allow for the reconstruction of vertical profiles of the measured trace gases and, using the oxygen collision complex $O_4$ as a proxy for the light path, also of aerosol extinction. Using suitable inverse models, trace gas and aerosol profiles can be retrieved in the lowermost $\approx$ 2 km with a vertical resolution of about 50 - 100 m at the surface, and a lower resolution above. Up to four independent pieces of information can be retrieved.

Algorithms for the retrieval of vertical profiles from MAX-DOAS measurements can be separated into those that retrieve vertical profiles on a finite vertical grid (usually with layers of 50 - 200 m thickness) using the optimal estimation method (OEM) (Rodgers, 2000), and parametrised algorithms that use a small number of parameters (typically 2-4) to describe the shape of the atmospheric profile. Parametrised algorithms are typically faster than OEM algorithms, since they are usually based on pre-calculated look-up tables (LUT), while OEM algorithms rely on on-line radiative transfer modelling (RTM). Being based on Bayesian statistics, OEM algorithms have the advantage of providing a thorough error analysis as well as a quantitative characterisation of the vertical resolution and the information content (Rodgers, 2000). However, the results of OEM algorithms critically depend on the appropriate choice of a priori constraints, which are in many cases difficult to assess. In addition to OEM and parametrised approaches, the present study also includes a fast algorithm developed by NASA, which relies only on geometrical considerations and only invokes radiative transfer modelling for a pure Rayleigh atmosphere.

Testing the performance of algorithms for the retrieval of the atmospheric state using remote sensing measurements on the basis of synthetic data is a method that has been widely used in the scientific community. In particular, numerous synthetic studies that investigated the performance of MAX-DOAS retrieval algorithms were published in the past (Wagner et al., 2004; Frieß et al., 2006; Hay, 2010; Vlemmix et al., 2011; Yilmaz, 2012; Hartl and Wenig, 2013; Holla, 2013; Zielcke, 2015). This paper presents the first intercomparison of eight state-of-the-art algorithms for the retrieval of vertical profiles of aerosols and trace gases using synthetic MAX-DOAS measurements. Synthetic measurements have the advantage over ambient measurements that the true atmospheric state is exactly known, and thus a quantitative comparison of true and retrieved atmospheric states is straightforward. This study is part of the Fiducial Reference Measurements for Ground-Based DOAS Air-Quality Observations ($FRM_4DOAS$) project funded by the European Space Agency (see http://frm4doas.aeronomie.be). One of the main objectives of this project is the development of a community algorithm for a harmonised near-real-time processing of MAX-DOAS data, including spectral analysis as well as the retrieval of tropospheric profiles of aerosols, HCHO and $NO_2$, and stratospheric $NO_2$ profiles as well as total ozone columns. As part of the $FRM_4DOAS$ project, the aim of the study presented here has been the selection of suitable algorithms for the retrieval of tropospheric profiles to be integrated in the $FRM_4DOAS$ community algorithm.

The paper is structured as follows: Section 2 briefly describes the inverse modelling theory and outlines the strategy for the intercomparison of the profile retrieval algorithms. A short description of the individual retrieval algorithms is provided in Section 3. The model scenarios and RTM settings are specified in Section 4. Slant column densities (SCDs) of $NO_2$, HCHO,

as well as the oxygen collision complex $O_4$, simulated by the different RTMs serving as forward models for the retrieval algorithms, are compared in Section 5. Comparisons of the quantities derived by the participating retrieval algorithms are presented in Section 6. These include averaging kernels from the OEM algorithms (Section 6.2), a posteriori modelled dSCDs (Section 6.3), vertical profiles (Section 6.4), total columns (Section 6.5), as well as aerosol extinctions and trace gas concentrations near the surface (Section 6.6). Finally, the numerical performance of the individual retrieval algorithms is assessed in Section 6.7.

## 2 Profile retrieval and intercomparison strategy

In general, the retrieval of the atmospheric state (or the state of any physical system) by remote sensing is based on the observation of a finite number of quantities that represent the components of the measurement vector $\mathbf{y}$, which is a function of the atmospheric state $\mathbf{x}$,

$$\mathbf{y} = \mathbf{F}(\mathbf{x}, \mathbf{b}) + \epsilon \tag{1}$$

Here, $\epsilon$ represents the measurement error. In case of MAX-DOAS retrievals, the state vector $\mathbf{x}$ consists either of aerosol extinction coefficients or trace gas concentrations in discrete atmospheric layers with a typical thickness of 50 - 200 m. The measured quantities are differential slant column densities (dSCDs) $dS$, usually the difference between the SCD $S$ at a certain EA $\alpha$ and the SCD from a zenith sky measurement, observed along different lines of sight:

$$dS = S(\alpha) - S(\alpha = 90°) \tag{2}$$

with the SCD $S$ being the integrated concentration along the (average) light path through the atmosphere.

$$S = \int \rho(s) \, ds \tag{3}$$

Here, $\rho$ is the number density of the trace gas and $s$ parametrises the light path length through the atmosphere. The vector $\mathbf{b}$ in Equation 1 represents additional forward model parameters, which are not to be retrieved, such as atmospheric pressure, temperature, aerosol optical properties, etc. The aim of an inverse model is to provide an estimate of the atmospheric state $\mathbf{x}$ for a given measurement $\mathbf{y}$. However, inverse problems are often poorly constrained, and the inverse of the forward model function $\mathbf{F}^{-1}$, for which $\mathbf{x} = \mathbf{F}^{-1}(\mathbf{y})$ either does not exist, or the finite measurement error $\epsilon$ leads to unstable estimates of the state vector.

To overcome these problems, MAX-DOAS retrieval algorithms make use of two different approaches. Retrieval algorithms using the well-known Optimal Estimation Method (OEM) are based on a Bayesian approach (Rodgers, 2000). They introduce

an a priori state vector $\mathbf{x_a}$, together with an a priori covariance matrix $\mathbf{S_a}$, as an additional constraint. OEM algorithms are based on the minimisation of the following cost function:

$$\chi^2(\mathbf{x}) = (\mathbf{y} - \mathbf{F}(\mathbf{x}, \mathbf{b}))^T \mathbf{S}_\epsilon^{-1} (\mathbf{y} - \mathbf{F}(\mathbf{x}, \mathbf{b})) + (\mathbf{x} - \mathbf{x_a})^T \mathbf{S_a}^{-1} (\mathbf{x} - \mathbf{x_a}) \tag{4}$$

Here $\mathbf{S}_\epsilon$ is the measurement covariance matrix, which, under the assumption that the measurements are independent, is a matrix with the squares of the measurement errors (specified in Section 6.1 and Table 6) as diagonal elements and zero values elsewhere. The most probable (maximum a posteriori, MAP) estimate $\hat{\mathbf{x}}$ is then given as

$$\hat{\mathbf{x}} = \arg\min \chi^2(\mathbf{x}) \tag{5}$$

The a priori constraints $\mathbf{x_a}$ and $\mathbf{S_a}$ represent the best knowledge of the atmospheric state before the measurement has been made, which can be derived, e.g., from climatologies, but are often only based on rough estimates of the typical atmospheric conditions at the measurement site. The averaging kernel matrix (AVK) $\mathbf{A}$ quantifies the sensitivity of the retrieved state to the true atmospheric state:

$$\mathbf{A} = \frac{\partial \hat{\mathbf{x}}}{\partial \mathbf{x}} \tag{6}$$

The degrees of freedom for signal (DFS) $d_s = Tr(\mathbf{A})$ quantify the number of independent pieces of information contained in the measurements. The $\mathbf{j}^{th}$ row of the AVK represents the sensitivity of the retrieved amount in the atmospheric layer $j$ to the amount in all other layers, and the retrieved profile can be expressed by the true atmospheric profile, smoothed by the AVK according to

$$\hat{\mathbf{x}} = \mathbf{x_a} + \mathbf{A}(\mathbf{x} - \mathbf{x_a}) \tag{7}$$

Parametrised retrieval algorithms do not explicitly introduce a priori constraints but overcome the problem that the state vector is poorly constrained by the measurements by representing the state vector as $\mathbf{x} = \mathbf{x}(\mathbf{p})$, using only a small number (typically 2-4) of parameters $\mathbf{p} = (p_1, \ldots, p_N)$ that describe quantities such as the total column, the layer thickness, and the shape of the profile. The cost function of parametrised algorithms is given as

$$\chi^2(\mathbf{p}) = (\mathbf{y} - \mathbf{F}(\mathbf{x}(\mathbf{p}), \mathbf{b}))^T \mathbf{S}_\epsilon^{-1} (\mathbf{y} - \mathbf{F}(\mathbf{x}(\mathbf{p}), \mathbf{b})) \tag{8}$$

and the best estimate of the parameters $\hat{\mathbf{p}}$ is given as

$$\hat{\mathbf{p}} = \arg\min \chi^2(\mathbf{p}) \tag{9}$$

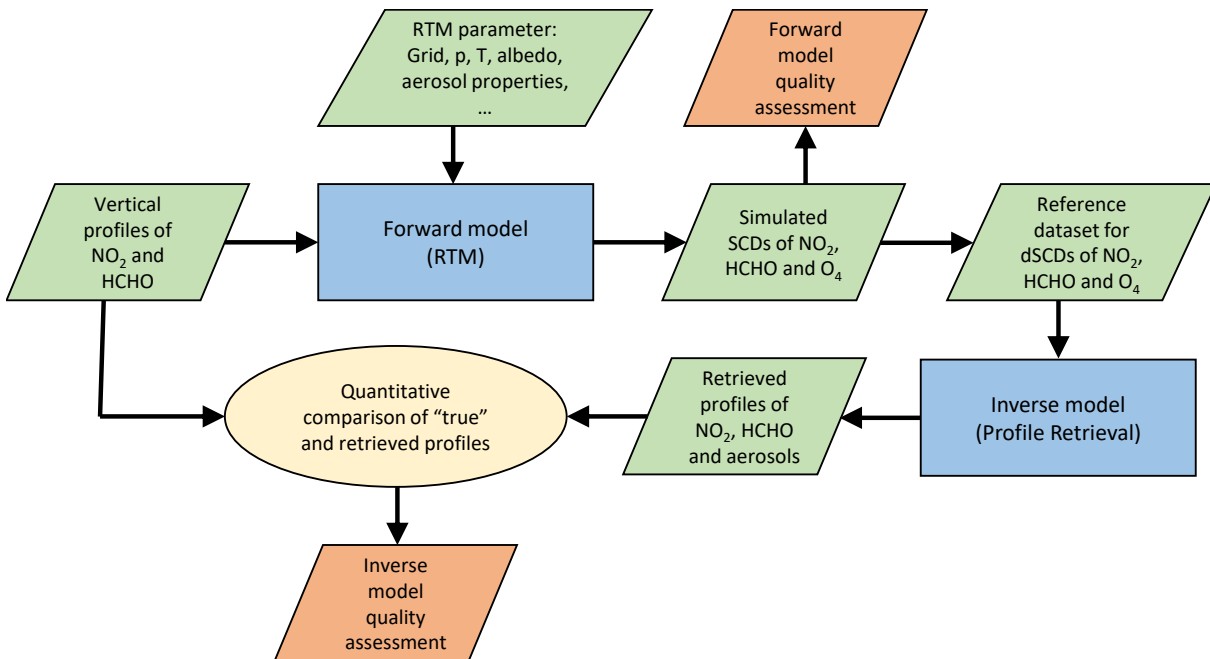

**Figure 1.** Flow diagram depicting the strategy for the retrieval algorithm intercomparison.

OEM algorithms have the advantages that the approach, based on the well-established Bayesian statistics, is mathematically stringent, and that important parameters, such as the retrieval covariance matrix (separated into smoothing and noise covariance), AVK, as well as information content, can be readily derived. Parametrised algorithms are usually faster than OEM algorithms, since the small number of parameters allows for the usage of pre-calculated LUTs, whereas OEM algorithms, with a larger number of state vector elements, usually perform radiative transfer calculations on-line. The calculation of the Jacobian matrix (weighting function), which is required by OEM algorithms for the minimisation of the cost function, can be quite time consuming, especially for non-linear problems, such as the aerosol profile retrieval. Parametrised algorithms have the disadvantage that the parametrisation limits the possible representations of the state vector to a certain subspace of the state vector space when characterising the state vector with a limited number of parameters. On the other hand, OEM algorithms tend to be biased to the a priori, in particular in regions where the sensitivity to the atmospheric state is low.

The overall strategy for the comparison of aerosol and trace gas vertical profiles from MAX-DOAS measurements within this study is depicted in Figure 1. It consists of the following steps:

1. The first step of the intercomparison exercise consists of a comparison of the forward models of the individual retrieval algorithms. Simulations of SCDs of HCHO, $NO_2$ and $O_4$ are performed for a variety of atmospheric scenarios and viewing geometries as described in Section 4. In the following, these prescribed scenarios are referred to as the true atmospheric state, which is represented by the atmospheric state vector $\mathbf{x}$. This forward model intercomparison exercise allows for a quantitative comparison of the individual RTMs on which the inversions are based (see Section 5).

**Table 1.** MAX-DOAS retrieval algorithms participating in this study

| Algorithm | Institute | Method | Forward Model |
|---|---|---|---|
| bePRO | BIRA-IASB | OEM[a] | LIDORT |
| BOREAS | IUP Bremen | OEM | SCIATRAN |
| HEIPRO | IUP Heidelberg | OEM(log)[b] | SCIATRAN |
| MMF | BIRA-IASB | OEM(log) | VLIDORT |
| PRIAM | MPIC | OEM(log) | SCIATRAN |
| MAPA | MPIC | PAR[c] | McArtim |
| MARK | KNMI | PAR | DAK |
| NASA | NASA | ANA[d] | N/A |

[a] Optimal estimation

[b] OEM(log): Optimal estimation with logarithmic state vector

[c] Parametrised retrieval

[d] Analytical retrieval with RTM only for Rayleigh atmosphere

2. The medians of the ensemble of dSCDs of HCHO, $NO_2$ and $O_4$ from the different forward models derived from step 1 serve as a reference dataset that represents the measurement vectors for the different atmospheric scenarios. This dataset was distributed among the participants of this study and serves as input for the individual retrieval algorithms. This dataset is referred to below as the (synthetic) measurements $\mathbf{y}$.

3. The comparison of profiles $\hat{\mathbf{x}}$ retrieved by each participant on the basis of the reference dataset with the true aerosol and trace gas profiles $\mathbf{x}$ allows for a quantitative assessment of the inverse models. The results of this intercomparison will be presented in Section 6.

4. Finally, a comparison of the numerical performance of the individual retrieval algorithms is performed (Section 6.7).

## 3 Retrieval algorithms

This section briefly describes the eight MAX-DOAS retrieval algorithms whose main features are listed in Table 1. Five of the algorithms are based on OEM, two on a parametrised approach, and one (NASA) on an analytical approach without radiative transfer modelling. Each of the participants is free to decide on criteria for flagging particular retrieval results as invalid as described in the following subsections. If not stated otherwise, the comparison of retrieved quantities within this study is based on all data, without considering the validity flags.

### 3.1 The bePRO algorithm

The Belgian Profile (bePRO) OEM inversion algorithm has been developed at the Royal Belgian Institute for Space Aeronomy (BIRA-IASB) by Clémer et al. (2010). The forward model used in bePRO is the linearised discrete ordinate radiative transfer

(LIDORT) code (Spurr et al., 2008). LIDORT is a multiple-scattering multi-layer discrete ordinate scattering code with a simultaneous linearization facility for the generation of both radiances and analytical Jacobians (intensity partial derivatives with respect to any atmospheric or surface parameter). It uses the Green's function method for solving the layer radiative transfer equations (RTEs), both for solar and thermal-emission sources. The analytical calculation of the weighting functions allows near-real-time MAX-DOAS profile inversion without the use of pre-calculated weighting function LUTs. bePRO uses a two-step approach. First, the aerosol extinction vertical profiles are retrieved for each MAX-DOAS scan from the corresponding measured $O_4$ dSCDs. Since it is a non-linear inversion problem, the iterative OEM approach is used to minimise the cost function defined in Equation 4. The standard aerosol retrieval output includes the following parameters: aerosol extinction profiles (in units of $km^{-1}$), aerosol optical depth, AVKs, smoothing and noise measurement error matrices, and $O_4$ dSCDs calculated from the retrieved aerosol extinction profiles using the forward model $\mathbf{F}$. In a second step, trace gas vertical profiles are retrieved using retrieved aerosol extinction profiles as input for the calculation of the corresponding weighting functions. Since trace gases (except ozone) are optically thin absorbers, the OEM equation for linear case is used (Rodgers, 2000).

The standard retrieval vertical grid used in bePRO is the following: ten layers of 200 m thickness starting from the altitude of the station, followed by two layers of 500 m thickness and one layer of 1 km thickness. This grid can be modified if needed.

A bePRO retrieval (aerosol or trace gas) is flagged as valid if the three following criteria are fulfilled: (1) the root mean square (RMS) of the difference between measured and simulated dSCDs < 30%, (2) DFS $\geq$ 1, and (3) no negative value in the retrieved profile.

## 3.2 The BOREAS algorithms

The Bremen Optimal estimation REtrieval for Aerosols and trace gaseS (BOREAS) OEM algorithm (Bösch et al., 2018) developed at the Institute of Environmental Physics (IUP) Bremen uses the Sciamachy radiative transfer model (SCIATRAN) version 4.0.1 for the forward model calculations and the SCIATRAN retrieval mode for the inversion of the aerosol extinction profile (Rozanov et al., 2014). However, since BOREAS is still under development, a previous version of SCIATRAN was used (v3.8.12) together with a preliminary version of BOREAS within this study. SCIATRAN is run in BOREAS using the discrete ordinate mode with full multiple scattering, full spherical geometry for single scattered and plane parallel geometry for multiple scattered light. Polarisation and rotational Raman scattering are usually not included in the forward model calculations. The weighting functions are calculated analytically assuming an optically thin atmosphere.

The BOREAS aerosol retrieval is based on the minimisation of the difference in $O_4$ optical depth between measurements and forward model within a Tikhonov-Regularization by varying the aerosol extinction profile until convergence or an iteration limit is reached. In a second step, so-called box air-mass factors are calculated for the chosen absorber using the afore obtained aerosol extinction profile. Again SCIATRAN in its full-spherical mode is applied. With the box air mass factors, the relation between the measurement and the absorber's profile in the atmosphere can be expressed as a linear system, which is then solved by applying the OEM on the measured slant columns and a priori profile information to retrieve trace gas profiles.

The profile retrieval is calculated based on SCIATRAN's main grid and can be set by the user. The grid itself describes homogeneous layers around the grid points, with the exception of the uppermost (lowermost) layer, which is considered as

half of the grid steps. BOREAS retrieval results are routinely calculated on equidistant grid levels from the surface up to the maximum retrieval height. For each level, the retrieved value is the average of the corresponding layer, e.g. in the case of the 200 m level, the altitude range from 100 m to 300m defines this specific layer. Two exceptions are the boundary levels, 0 m and 4000 m, which cover only half the altitude range compared to the other layers (0 - 100 m and 3900 m - 4000 m). Since the grid was defined at the centre of the levels in this intercomparison study, BOREAS results were interpolated on these altitude values. Due to this interpolation, the lowest values in the submitted profiles are the interpolation between the retrieved surface value and the 200 m result.

Four different quality filters were applied. A profile is flagged as invalid if either (1) the retrieved vertical column density is negative, (2) the profile contains more than ten negative values, (3) the RMS between simulated and measured dSCD is larger than $2 \cdot 10^{16}$ molec/cm$^2$ for NO$_2$ and HCHO, and $5 \cdot 10^{43}$ molec$^2$/cm$^5$ for O$_4$, and (4) the relative difference between the simulated and the measured dSCD for each line of sight is smaller than 5%.

### 3.3 The HEIPRO algorithm

The Heidelberg Profile Retrieval Algorithm (HEIPRO) is an updated version of the algorithm already described in detail in Frieß et al. (2006) and Frieß et al. (2011). It uses the SCIATRAN radiative transfer model version 2.1.5 (Rozanov et al., 2014) in discrete ordinate mode with full multiple scattering, full spherical geometry for single scattered and plane parallel geometry for multiple scattered light. The weighting functions for trace gases (box airmass factors) are calculated analytically, the weighting functions for aerosols are calculated using the finite difference method.

HEIPRO is based on OEM and retrieves the most probable state vector by minimizing the cost function given by Equation 4. The vertical extinction and aerosol profiles are represented in the state vector as the logarithm of extinction and trace gas concentration, respectively. This has the advantages that (1) negative values, which cannot be processed by the RTM, are avoided, (2) the retrieval can reconstruct a larger range of atmospheric conditions, and (3) the retrieval is generally more stable with less tendency to oscillations. As most MAX-DOAS retrieval algorithms, HEIPRO uses a two-step approach, where the aerosol extinction profile is retrieved in a first step using O$_4$ dSCDs as measurement vector, and the trace gas profile is retrieved in a second step using the according trace gas dSCDs as measurements vector, together with the aerosol profile from the first step. The radiative transfer model SCIATRAN version 2.1.5 serves as forward model for the retrieval.

No filtering of the HEIPRO data has been performed, and all profiles are flagged as valid.

### 3.4 The MMF algorithm

The Mexican Maxdoas Fit (MMF) algorithm (Friedrich et al., 2018) uses VLIDORT (Spurr, 2006) version 2.7, the vectorized version of LIDORT (see also Sect. 3.1), in pseudo-spherical multiple scattering setting as forward model. Only the intensity information and its analytically calculated Jacobians, not the other Stokes parameters, are currently used. MMF can operate in linear or logarithmic state vector space. While different regularization matrices are possible in the linear space, only $S_a$ matrices of the form used in this study are currently tested in logarithmic mode. The results presented here were performed in logarithmic mode. MMF also uses a two-step approach, as outlined above in Sect. 3.3. Quantities such as aerosol single

scattering albedo, aerosol asymmetry factor, surface albedo, as well as temperature and pressure profiles can be supplied. On top of the custom retrieval grid, a simulation grid can be supplied. The algorithm has so far been applied in linear mode with Tihkonov regularization for aerosol retrieval on an almost equal distance-in-pressure grid at UNAM (National Autonomous University of Mexico) for the retrieval of $NO_2$ profiles from 4 stations in the Mexico City area.

MMF flagging for this study was based on the mean of the ratio of the absolute value of the difference between measured and simulated dSCD and the dSCD measurement error. The limit for flagging scans as invalid was 10.

### 3.5    The PRIAM algorithm

The OEM-based profile inversion algorithm of aerosol extinction and trace gas concentration (PriAM) developed by the Anhui Institute of Optics and Fine Mechanics, Chinese Academy of Sciences (AIOFM, CAS), in cooperation with Max Planck

Institute for Chemistry (MPIC), is introduced in Wang et al. (2013) and Frieß et al. (2016). A two-step inversion procedure is used in PriAM. In the first step, profiles of aerosol extinction are retrieved from the $O_4$ dSCDs. The single scattering albedo and asymmetry factor should be defined for the aerosol retrieval based on other auxiliary measurements. Afterwards, profiles of volume mixing ratios (VMRs) of trace gases are retrieved from the respective dSCDs in each MAX-DOAS EA sequence. The retrieval problem is solved by the Levenberg-Marquardt modified Gauss-Newton numerical iteration procedure

(Levenberg, 1944; Marquardt, 1963; Rodgers, 2000). PriAM uses the SCIATRAN 2.2 RTM (Rozanov et al., 2005) to calculate the weighting function and other simulated quantities. To avoid meaningless negative values, the original aerosol extinction and VMR of trace gases are transformed to the logarithms of these quantities. Because of the conversion, it is necessary to use the nonlinear optimal inverse method to retrieve the profiles of trace gases instead of the linear method. PriAM can retrieve trace gas and aerosol profiles on any arbitrary vertical grid.

### 3.6    The MAPA algorithm

The Mainz Profile Algorithm (MAPA) developed at the MPIC is a two-step algorithm based on a parametrised approach. First, the aerosol profile is retrieved based on $O_4$ DSCDs, and second, trace gas profiles are retrieved based on trace gas dSCDs and aerosol profiles from the first step. The forward model is provided as a LUT, relating the profile parameters to $O_4$ and trace gas differential airmass factors (dAMFs) for given solar zenith angles (SZA) and relative azimuth angles (RAA) between the

Sun and the instrument's field of view. The LUT is calculated with the Monte Carlo Atmospheric Radiative Transfer Inversion Model (McARTIM) (v1), a full spherical Monte-Carlo model without polarisation (Deutschmann et al., 2011). Up to four parameters are determined independently: the integrated column (aerosol optical thickness, AOT; or vertical column density, VCD), layer height, profile shape, and, in case of aerosols, the $O_4$ scaling factor (optional). The profile shape is described by the shape parameter $s$, with $s = 1$ representing a box profile, $s < 1$ representing a combined box profile with an exponential

profile on top ($s$ describes the fraction of the box profile on the total profile), and $s > 1$ representing elevated profiles.

     Previous versions of the parameter-based profile inversions, as described e.g. in Wagner et al. (2011), were based on a Levenberg-Marquardt least-squares algorithm. In MAPA (from v0.6 on), however, the profile retrieval is based on a Monte Carlo approach yielding an ensemble (instead of only one set) of best matching parameters. This approach is much faster

(about 3 seconds per profile), accounts for correlations between the parameters, can deal with multiple minima, and allows the uncertainty of the resulting profiles to be determined. MAPA is described in detail in Beirle et al. (2018).

MAPA uses RTM parameters for the LUT generation that slightly differ from those prescribed within this study (see Section 2), with a phase function asymmetry parameter of 0.68, aerosol single scattering albedo of 0.95, and a surface albedo of 0.05.

The MAPA flagging scheme is as follows: For each elevation sequence, MAPA determines the parameter combinations yielding best match of modeled and measured dSCDs, no matter how good this "best match" actually is. Thus, flagging is required in order to evaluate which MAPA results should be considered as meaningful and which not. MAPA provides a 2-stage flagging scheme: moderate exceedance of the thresholds results in a warning, while large deviations raise an error. Flagging is based on different criteria: (1) The level of agreement between forward model and measurement as compared to

the dSCD error (for details see Beirle et al., 2018), (2) the consistency of column parameter within the Monte Carlo ensemble, and (3) the shape of the resulting profile, which is requested to be located in the lower troposphere. In addition, scenes with high AOT ($> 2$) are flagged as invalid. For RAA $< 15°$, even scenes with AOT $> 0.5$ are flagged. All flags are determined and stored individually, and merged into one total flag defined as any of the flag criteria for both warnings and errors.

Flag criteria and thresholds have been developed and optimized based on both the synthetic dSCDs presented in this study

and real measurements during the Second Cabauw Intercomparison of Nitrogen Dioxide Measuring Instruments (CINDI-2). For details on the MAPA flagging scheme and a discussion on the impact of the a priori thresholds see Beirle et al. (2018).

## 3.7   The MARK algorithm

The MAXDOAS retrieval KNMI (MARK) developed at the Royal Dutch Meteorological Institute (KNMI) is described in Vlemmix et al. (2011, 2015a). It makes use of a profile shape parametrisation with just a few (2-4) free parameters. A LUT

of differential slant column simulations is produced by the Doubling Adding model KNMI (DAK) (de Haan et al., 1987; Stammes et al., 1989). A standard least-squares algorithm is used to minimise the deviations between simulated and measured differential slant columns at the different EAs. Uncertainties to the parameters are estimated from the spread in the results of an ensemble approach in which the retrieval is performed multiple times with disturbed measured differential slant column densities, based on the DOAS retrieval uncertainties. For each individual retrieval the aerosol profile is retrieved first, and the

outcome is used in the trace gas retrieval. The ensemble retrieval is done for four different parameterizations, and the reduced $\chi^2$ distribution of the ensembles is used to make an a posteriori composite of the profile and its corresponding uncertainties. The fitted profile shape parameters are: (1) the tropospheric column density (or AOT); (2) the top height of the mixing layer; (3) the "shape parameter", which determines the linear increase or decrease in the mixing layer; (4) the fraction of the total trace gas column density which resides above the mixing layer.

MARK data are flagged as invalid if the variability in the AOT or the TG-column within an ensemble is larger than 15% of the value itself.

## 3.8 The NASA algorithm

The National Aeronautics and Space Administration (NASA) Real Time algorithm is developed as a quick look algorithm that relies on the fact that atmospheric scattering strongly affects DOAS measured $O_4$ absorption (Spinei et al., Fast aerosol extinction coefficient profile estimation from MAXDOAS UV/VIS measurements, in preparation). Two separate approaches are
5 used for aerosol and trace gas profile retrieval. The aerosol profile algorithm determines the layer aerosol extinction coefficients by comparing measured Ring and $O_4$ absorption with Ring and $O_4$ absorption under pure Rayleigh conditions. Air mass factors and Ring absorption for the Rayleigh case are pre-calculated using the VLIDORT v2.8 and LIDORT-LRRS v2.5 radiative transfer models, respectively (Spurr et al., 2008; Spurr, 2008) assuming the U.S. standard atmosphere. Since Ring simulations were not provided in this study, aerosol analysis was performed only at 477 nm. $O_4$ dSCDs are corrected for SZA dependence.
Eq. 10 is the simplified equation used in this study to calculate aerosol scattering extinction coefficients at each layer for specific observation geometry (EA and RAA) $\Theta$. We also assume an aerosol single scattering albedo of $\omega_{aer}(\lambda) = 1$.

$$\epsilon_{aer}(\lambda, \Theta, \vartheta) \approx \frac{\tau_{O4,i}^{noaer} - \tau_{O4,i}^{aer}}{\Delta h} \tag{10}$$

with $\tau_{O4,i}^{aer}$ and $\tau_{O4,i}^{noaer}$ being the slant optical density with and without aerosols in the respective layer $i$, $\lambda$ denoting wavelength, and $\vartheta$ the SZA. The thickness $\Delta h$ of the respective layer is determined from the $O_4$ dSCDs using simple trigonometry
according to Eq. 11 and 12, resulting in an atmosphere specific grid:

$$h_i = \frac{\Delta S_{O4}(\alpha_i) + V_{O4}}{n_{O4}} \cdot \sin \alpha_i \tag{11}$$

$$\Delta h = h_{i+1} - h_i \tag{12}$$

with $S_{O4}(\alpha_i)$ being the $O_4$ slant column density at elevation angle $\alpha_i$ and $V_{O4}$ the $O_4$ vertical column density.

The maximum number of vertical layers is equal to the number of elevation angles. Profiles are considered invalid if less
than four measurements are used in the profile calculation, with all of the synthetic data analysed here satisfying this test. Within this study, an exponential profile reducing to 0.01% of the last altitude layer extinction coefficient at 4 km was added for consistency with the other algorithms. The resulting profile was then linearly interpolated on the common grid (200 m up to 4 km).

The trace gas profile retrieval does not rely on the aerosol retrieval. The trace gas VCD $V_{gas}$ is calculated first from the trace
gas and $O_4$ dSCD measurements at 15° EA, $\Delta S_{gas}^{15°-90°}$:

$$V_{gas} = \frac{\Delta S_{gas}^{15°-90°}}{\Delta A_{O4}^{15°-90°}} \tag{13}$$

with the according $O_4$ dAMF $\Delta A_{\text{gas}}^{15°-90°}$ calculated via

$$A_{\text{O4}}^{15°-90°} = \frac{\Delta S_{\text{O4}}^{15°-90°}}{V_{\text{O4}}} + 1 \qquad (14)$$

Near-surface trace gas VMR $M_{\text{gas}}$ are calculated by simple extrapolation of trace gas and $O_4$ dSCDs at 1° and 2° to 0° EA, yielding $\Delta S(\alpha, \lambda)_{\text{gas}}^{\text{extrapolated}}$ and $\Delta S(\alpha, \lambda)_{\text{O4}}^{\text{extrapolated}}$, and by converting to the VMR using surface pressure $p$ and temperature $T$ similar to Sinreich et al. (2013):

$$M_{\text{gas}} = \frac{\Delta S(\alpha, \lambda)_{\text{gas}}^{\text{extrapolated}}}{\Delta S(\alpha, \lambda)_{\text{O4}}^{\text{extrapolated}}} \cdot \frac{p \cdot N_a}{R \cdot T} \cdot \chi_{\text{O2}}^2 \qquad (15)$$

The rest of the profile VMR is calculated using $O_4$ and trace gas dSCDs at multiple EAs. The layer altitude is calculated similar to the aerosol case. The derived profile is then converted to partial columns and scaled by the total VCD. As in the aerosol case, the layer grid is condition specific and was adjusted to the common grid in this study.

## 4 Model scenarios and RTM settings

A first important step for the comparison of retrieval algorithms is the assessment of their capability to realistically simulate the underlying physical processes using appropriate forward models, which are in this case atmospheric radiative transfer models. This section describes the forward model parameters and atmospheric scenarios for the modelling of $O_4$, $NO_2$ and HCHO SCDs, while the comparison of forward modelled SCDs is presented in Section 5.

The model atmosphere for the forward calculations consists of 67 layers, with a resolution of 100 m at altitudes between the surface and 4 km, and a coarser resolution above. Note that the retrieval of extinction and trace gas profiles is performed on a coarser grid with 200 m resolution in the lowermost 4 km. The choice of a finer grid for the forward modelling than for the inverse modelling allows for the investigation of the impact of sub-grid trace gas and aerosol variabilities on the retrieved profiles. For the forward modelling, a constant concentration within each layer has been implemented for the model calculations whenever possible.

The trace gas concentration and aerosol extinction profile scenarios for the forward modelling of HCHO, $NO_2$ and $O_4$ SCDs are shown in Figure 2. The same set of trace gas profiles is assumed for both HCHO and $NO_2$, in accordance with ambient measurements in mid-latitudes, where typical concentrations of both species are of the same order of magnitude (e.g., Vlemmix et al., 2015b). For simplicity, it is assumed that aerosol extinction does not change with wavelength (Ångström exponent of zero). The model profiles are chosen in order to represent a large variety of different atmospheric conditions, including trace-gas and aerosol free atmospheres as well as moderate to high trace gas and aerosol loads up to cloudy and foggy conditions. The shapes of the different profiles include near-surface box profiles, profiles exponentially decreasing with altitude, uplifted profiles with a Gaussian shape, as well as profiles without trace gases and aerosols, respectively. Additionally, the trace gas scenarios include two $NO_2$ profiles measured with a balloon sonde during the CINDI-2 campaign held in the Netherlands in

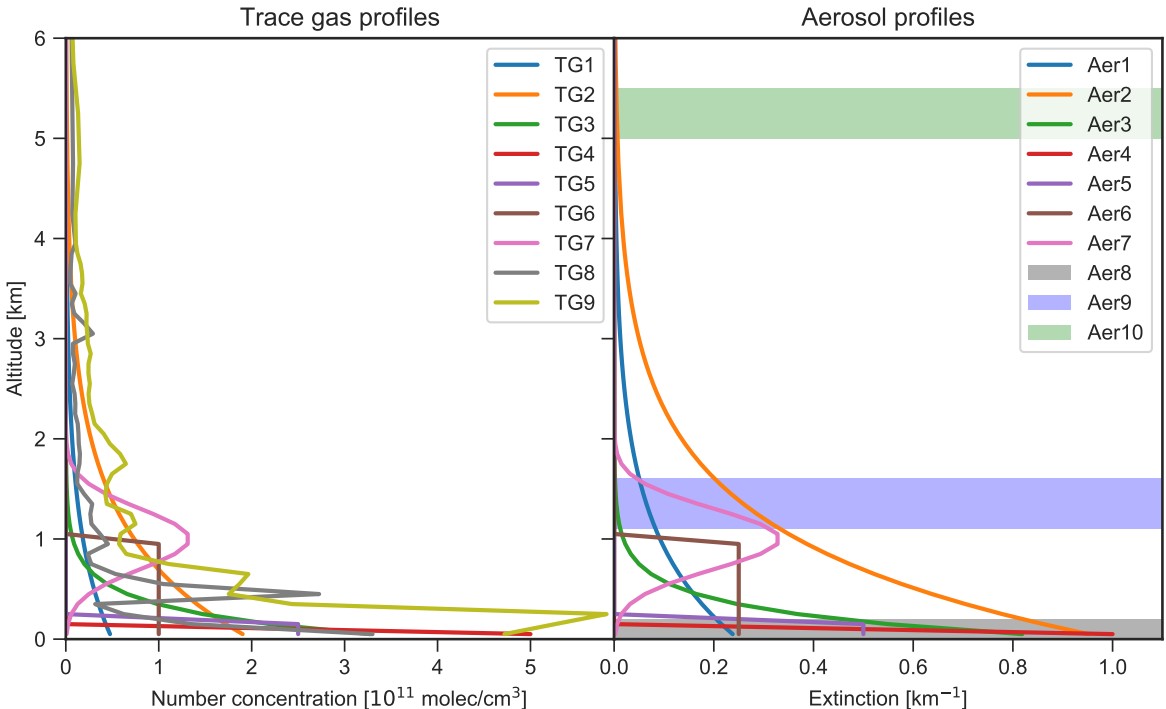

**Figure 2.** Trace gas number concentration (left) and aerosol extinction (right) vertical profiles as input for the forward modelling of HCHO, $NO_2$ and $O_4$. The shaded areas in the right panel indicate the location of fog and cloud layers with an extinction coefficient of $10\ km^{-1}$. The properties of the individual profiles are described in Tables 2 and 3.

September 2016 (Kreher et al., Intercomparison of $NO_2$, $O_4$, $O_3$ and HCHO slant column measurements by MAX-DOAS and zenith-sky UV-Visible spectrometers, in preparation for AMTD, see also http://www.tropomi.eu/data-products/cindi-2). The aerosol scenarios include three extreme cases with an extinction of $10\ km^{-1}$ located at the surface, as well as at altitudes around 1.3 km and 5.25 km, representing fog, low-lying and high-lying clouds, respectively. In total, the scenarios include 10 trace gas and 11 aerosol profiles whose features are listed in Tables 2 and 3, respectively. The forward calculations of $O_4$ at 360 nm and 477 nm are performed for each of the 11 aerosol profiles. $NO_2$ and HCHO SCDs are simulated for each combination of aerosol and trace gas profiles, yielding in total 110 different trace gas scenarios.

The viewing geometry for the forward model simulations is specified by the EA $\alpha$, the SZA $\theta$ and the RAA $\phi$ between the Sun and the viewing direction of the instrument. Simulations are performed using any combination of values for EA, SZA and RAA as listed in Table 4. The EA sequence, consisting of 10 angles between the horizon and zenith, is identical to the measurement sequence during the CINDI-2 campaign. Together with three SZA and three RAA values, this yields 90 different viewing geometries to be simulated for each model atmosphere.

**Table 2.** Description of the trace gas profiles shown in Figure 2. The VCD and the maximum number concentration $\rho_{max}$ are given in units of $10^{15}$ molec/cm$^2$ and $10^{11}$ molec/cm$^3$, respectively.

|  | Description | VCD | $\rho_{max}$ |
|---|---|---|---|
| TG0 | No trace gas | 0.00 | 0.00 |
| TG1 | Exponential, 1 km scale height | 5.00 | 0.48 |
| TG2 | Exponential, 1 km scale height | 19.99 | 1.90 |
| TG3 | Exponential, 250 m scale height | 9.93 | 3.27 |
| TG4 | Box profile, 100 m height | 5.00 | 5.00 |
| TG5 | Box profile, 200 m height | 5.00 | 2.50 |
| TG6 | Box profile, 1 km height | 10.00 | 1.00 |
| TG7 | Gaussian at 1 km, 300 m FWHM | 10.00 | 1.31 |
| TG8 | $NO_2$ balloon sonde (CINDI-2, 20160914) | 17.73 | 3.30 |
| TG9 | $NO_2$ balloon sonde (CINDI-2, 20160921) | 40.88 | 5.82 |

**Table 3.** Description of the aerosol extinction profiles shown in Figure 2. The maximum extinction $k_{max}$ is given in units of km$^{-1}$.

|  | Description | AOT | $k_{max}$ |
|---|---|---|---|
| AER0 | No aerosols | 0.00 | 0.00 |
| AER1 | Exponential, 1 km scale height | 0.25 | 0.24 |
| AER2 | Exponential, 1 km scale height | 1.00 | 0.95 |
| AER3 | Exponential, 250 m scale height | 0.25 | 0.82 |
| AER4 | Box profile, 100 m height | 0.10 | 1.00 |
| AER5 | Box profile, 200 m height | 0.10 | 0.50 |
| AER6 | Box profile, 1 km height | 0.25 | 0.25 |
| AER7 | Gaussian at 1 km, 300 m FWHM | 0.25 | 0.33 |
| AER8 | Box profile, 200 m height (fog) | 2.00 | 10.00 |
| AER9 | Cloud between 1.1 km and 1.6 km | 5.00 | 10.00 |
| AER10 | Cloud between 5.0 km and 5.5 km | 5.00 | 10.00 |

Further RTM parameters specified for both forward and inverse modelling are listed in Table 5, and the wavelengths for the simulation of the different trace gases can be found in Table 6. The RTM parameters include vertical profiles of temperature and pressure derived from average ozone sondes for the month of September from 2013 to 2015 in De Bilt, the Netherlands, as well as surface albedo, aerosol optical properties and trace gas literature absorption cross sections. The aerosol scattering was parametrised using a Henyey-Greenstein phase function (Henyey and Greenstein, 1941) with the asymmetry parameter as listed in Table 5.

**Table 4.** Viewing geometry for the model calculations. SCDs are simulated for each combination of EA, SZA and RAA.

| Parameter | Values |
|---|---|
| EA $\alpha$ | 1°, 2°, 3°, 4°, 5°, 6°, 8°, 15°, 30°, 90° |
| SZA $\theta$ | 40°, 60°, 80° |
| RAA $\phi$ | 0°, 90°, 180° |

**Table 5.** RTM parameters for the radiative transfer modelling.

| Parameter | Value |
|---|---|
| Temperature and pressure profile | Sept. 2013-2015 average from De Bilt ozone sondes |
| Surface albedo (Lambertian surface) | 0.06 |
| Aerosol single scattering albedo | 0.92 |
| Aerosol phase function asymmetry parameter | 0.68 |
| Instrument height | 0 km |
| $O_4$ absorption cross section | 293 K (Thalman and Volkamer, 2013) |
| $NO_2$ absorption cross section | 298 K (Vandaele et al., 1998) |
| HCHO absorption cross section | 297 K (Meller and Moortgat, 2000) |

In total, all combinations of viewing geometries, aerosol profiles and trace gas profiles yield 990 $O_4$ SCD simulations each at 360 nm and 477 nm, and 9900 simulations for each of the trace gases $NO_2$ and HCHO.

## 5   Intercomparison of forward modelled SCDs

In this section, the ability of the forward models to realistically simulate trace gas SCDs is assessed based on a comparison of the individual simulations with the median of the SCDs from the different RTMs. The median SCDs also serve as a reference dataset and provide the synthetic measurements for the profile retrieval comparison presented in Section 6. Figure 3 shows the comparison of the simulated $O_4$, $NO_2$ and HCHO SCDs to the respective ensemble median. The respective parameters of a linear regression of forward modelled SCDs versus median SCDs are shown in Figure 4. The SCDs simulated by most forward models agree under all conditions, with Pearson's coefficients and slopes very close to unity ($R > 0.999$ and $0.99 < \text{slope} < 1.001$ ) for a regression between SCDs from the individual models and the median from all models. Exceptions are HEIPRO and PRIAM, both using an older version of the SCIATRAN RTM (version 2.1) as the forward model. These models show deviations for the fog and clouds scenarios (AER8 ... AER10), the shallow box profile (AER4), and in the case of HEIPRO also for the exponential profile AER2 with a high AOT of 1. MAPA with McArtim as forward model yields significantly smaller slopes of 0.955 and 0.959 than the other models for $O_4$ at 477 nm and for $NO_2$, respectively. This is probably due to the different treatment of sphericity within the McArtim model. Significant biases are found for MAPA/McArtim and

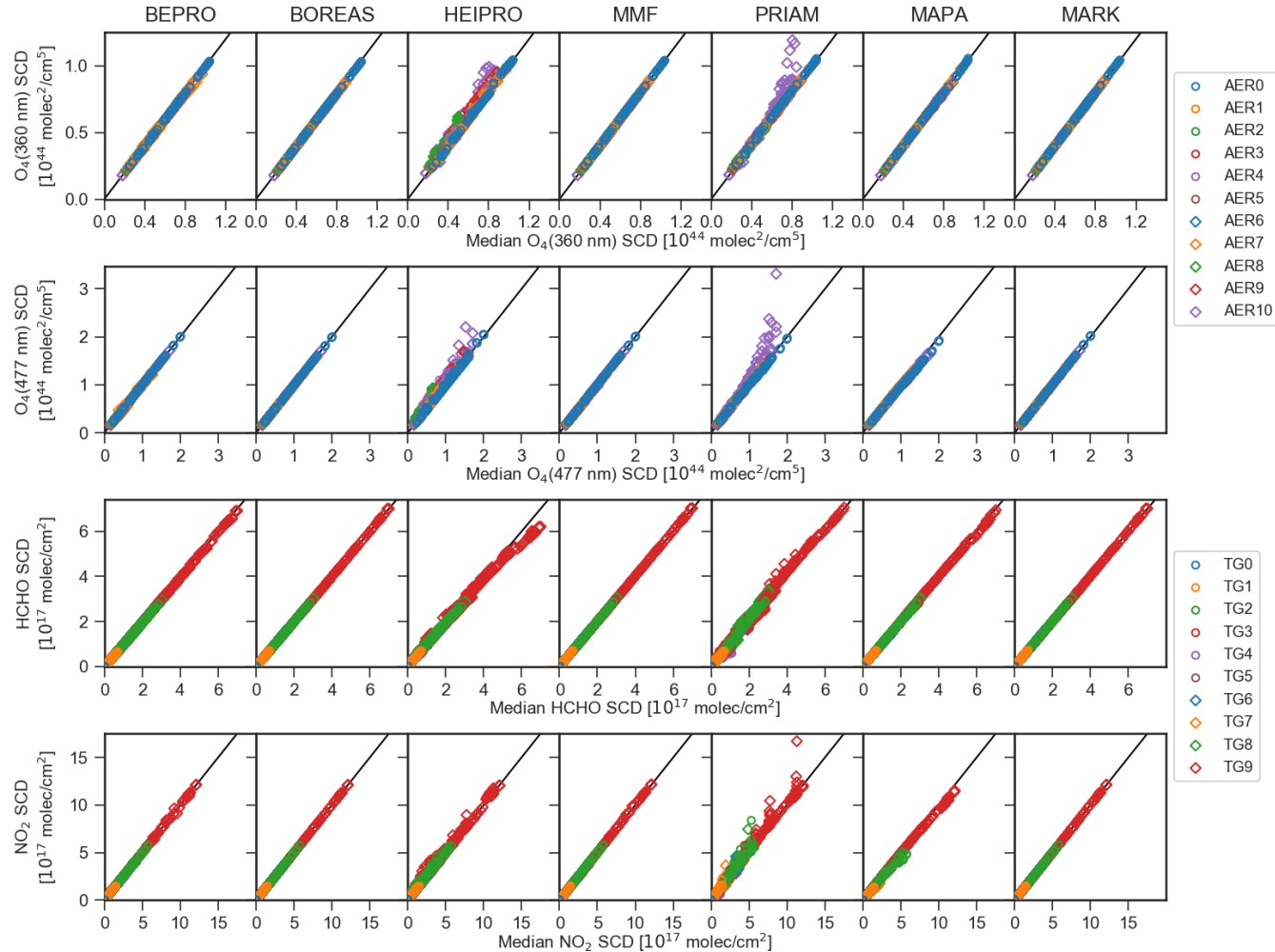

**Figure 3.** Correlation of SCDs of $O_4$ (360 and 477 nm), HCHO and $NO_2$ simulated by the different forward models with the corresponding median SCDs from all models. Colours indicate the underlying aerosol and trace gas profiles, respectively, as denoted in the legend. For HCHO and $NO_2$, results for all scenarios with different aerosol profiles are included.

PRIAM/SCIATRAN for $NO_2$ and $O_4$ at 477 nm, as well as for HEIPRO/SCIATRAN for HCHO. Apart from possible differences in the implementation and the approaches of the individual RTMs, some of the differences are likely due to the different representations of the trace gas and aerosol profiles on the prescribed vertical grid.

In summary, the SCDs simulated by the different radiative transfer models agree well under all conditions, as within previous
5   RTM intercomparisons (e.g., Wagner et al., 2007). This implies that possible discrepancies in the retrieved trace gas and aerosol profiles are mainly caused by differences in the implementation of the retrieval algorithms, rather than by differences in the forward models.

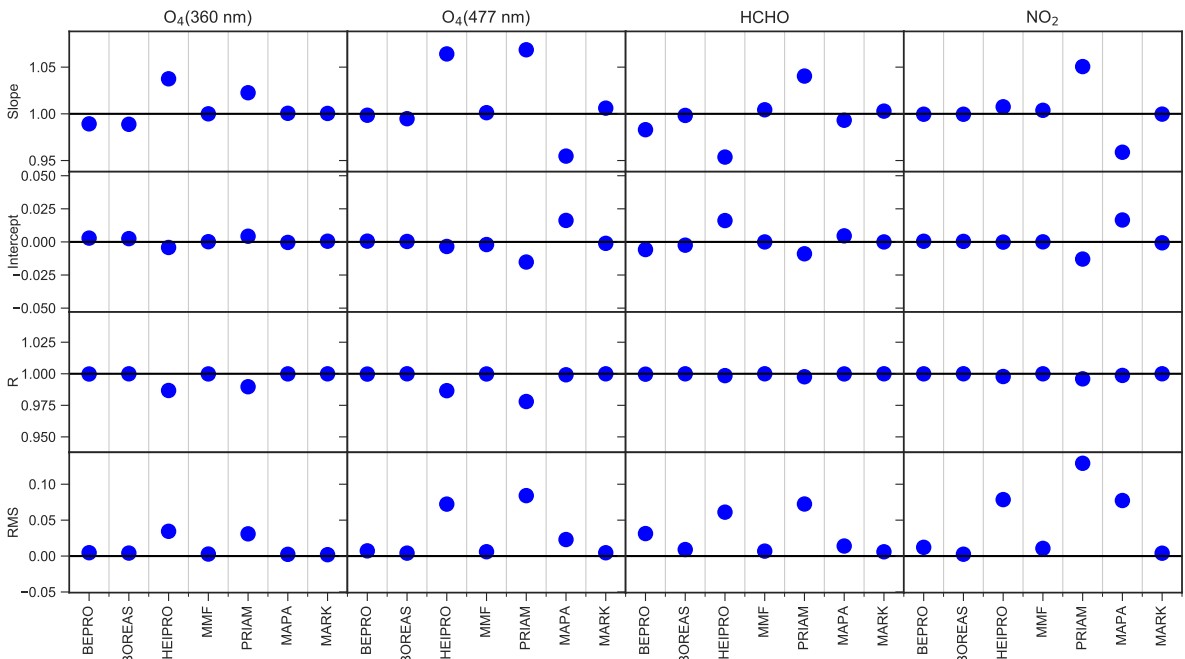

**Figure 4.** Slope, intercept, regression coefficient and root mean square difference (RMS) of the correlation between the SCDs from the individual forward models and the median SCDs from all models. RMS and intercept values are in units of $10^{43}$ molec$^2$/cm$^5$ for $O_4$, and $10^{17}$ molec/cm$^2$ for HCHO and $NO_2$.

## 6 Retrieval algorithm comparison

This section presents the results of the aerosol and trace gas retrieval algorithm intercomparison. Section 6.1 describes the measurement vectors, forward model parameters and a priori constraints of the profile retrieval. An important diagnostic tool is provided by the AVKs from the OEM algorithms, which quantify the sensitivity of the retrieval to the atmospheric state. AVKs and information content are discussed in Section 6.2. The comparison of modelled and measured dSCDs presented in Section 6.3 provides a measure for the level of convergence of the retrieval algorithms. Results of the intercomparison of vertical profiles, trace gas and aerosol total columns, as well as surface extinction and trace gas concentrations are presented in Sections 6.4, 6.5, and 6.6, respectively. Finally, the computational speed of the retrieval algorithms is discussed in Section 6.7.

### 6.1 Retrieval settings

Based on the simulated SCDs from the atmospheric scenarios described in Section 5, a reference dataset of $O_4$ at 360 nm and 477 nm, $NO_2$ and HCHO dSCDs has been compiled based on the ensemble median dSCDs from all participants. In the following, we refer to this synthetic dataset as the measured dSCDs. Table 6 lists the according dSCD errors, which were chosen to represent typical values based on ambient measurements by various instruments during the CINDI-2 campaign. As

**Table 6.** Assumed measurement errors of the $O_4$, HCHO and $NO_2$ dSCDs, as well as total columns of the respective a priori profiles. Units for the errors are $molec^2/cm^5$ for $O_4$ and $molec/cm^2$ for HCHO and $NO_2$.

| Species | dSCD error | $\lambda$ [nm] | A priori total column |
|---------|-----------|----------------|----------------------|
| $O_4$ | $2 \cdot 10^{41}$ $molec^2/cm^5$ | 360 | 0.18 |
| $O_4$ | $2 \cdot 10^{41}$ $molec^2/cm^5$ | 477 | 0.18 |
| HCHO | $2 \cdot 10^{15}$ $molec/cm^2$ | 343 | $8 \cdot 10^{15}$ $molec/cm^2$ |
| $NO_2$ | $5 \cdot 10^{14}$ $molec/cm^2$ | 460 | $9 \cdot 10^{15}$ $molec/cm^2$ |

already discussed in Section 4, the reference dataset contains dSCDs at 9 EAs for 990 different combinations of trace gas profiles, aerosol profiles and solar geometry (SZA and RAA).

Two reference datasets were created: Dataset v1 contains the median dSCDs without any noise, and dataset v1n contains the median dSCDs with a noise component consisting of the sum of (1) normally distributed noise with a standard deviation according to the errors listed in Table 6, and (2) additional noise with a standard deviation of 5% of the dSCD values. This second noise component, which is in case of $O_4$ and $NO_2$ much higher than typical measurement noise, is supposed to represent random variations in the measured dSCDs due to horizontal inhomogeneities of aerosols and trace gases as well as temporal changes in the atmospheric state (in particular cloud properties) during a single elevation scan with a typical duration of several minutes. The second noise component is of similar magnitude as the RMS differences between dSCDs measured by more than 30 instruments during the CINDI-2 Campaign (Kreher et al., Intercomparison of $NO_2$, $O_4$, $O_3$ and HCHO slant column measurements by MAX-DOAS and zenith-sky UV-Visible spectrometers, in preparation for AMTD).

Each participant performed retrievals with settings being as close as possible to the prescribed settings described below. The results based on noise-free and noisy dSCDs are labelled 'v1' and 'v1n', respectively. Each measurement vector consists of $O_4$, HCHO or $NO_2$ dSCDs of a single EA sequence for a given atmospheric scenario specified by aerosol and trace gas profiles, as well as solar zenith and relative azimuth angles. Retrievals are performed using the same forward model parameters as for the forward model calculations (see Table 5), with the exception of MAPA that uses slightly different surface albedo and aerosol optical properties (see Section 3.6). The retrieval grid consists of 20 layers with 200 m thickness between the surface and 4 km altitude, which means that the atmospheric layers are twice as thick as for the forward modelling. This choice allows for the investigation of the impact of sub-grid variations on the retrieval, in particular in case of scenarios TG4 and AER4 (100 m thick box profiles).

In contrast to the output of OEM algorithms, which directly retrieve trace gas and aerosol profiles on this vertical grid, profiles from the parametrised algorithms are interpolated onto the prescribed grid. The OEM algorithms use aerosol and trace gas a priori profiles exponentially decreasing with altitude with a scale height of 1 km, with a priori vertical columns for each species as listed in Table 6. The diagonal elements of the a priori covariance matrices consist of the square of 50% of the corresponding a priori profiles. Off-diagonal terms consist of Gaussian function with a correlation length of 200 m as described by Clémer et al. (2010).

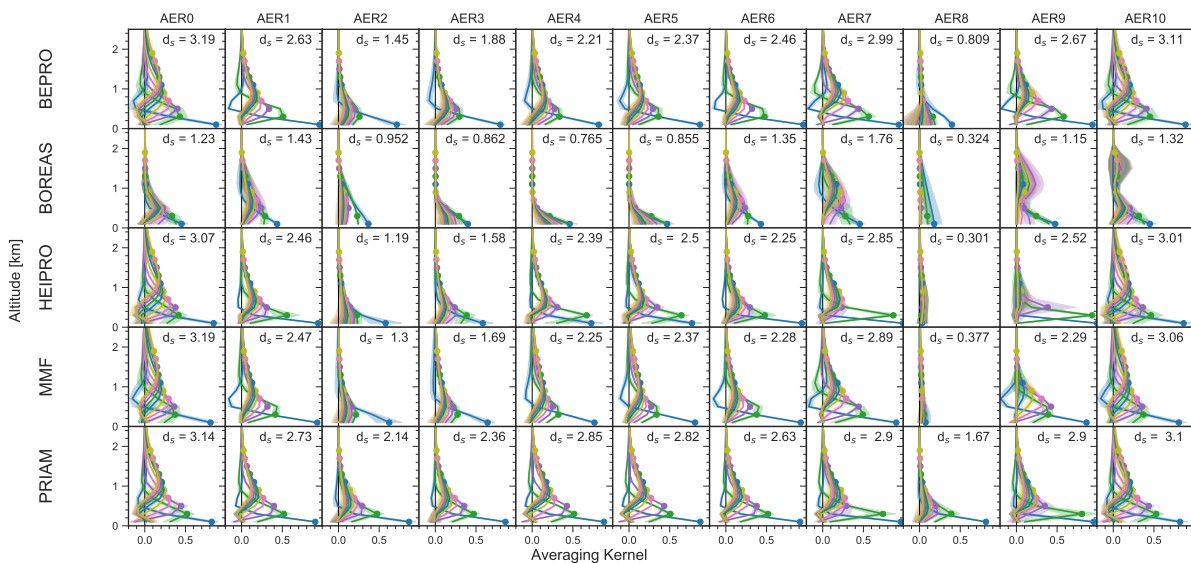

**Figure 5.** 360 nm aerosol AVKs. Each subplot shows the mean AVK over all SZA and RAA for a specific aerosol scenario (columns) and retrieval algorithm (rows). Filled circles indicate the nominal altitude of the corresponding AVK plotted in the same colour. Colour shaded areas (barely visible in most cases) indicate the standard deviation of the AVKs, i.e. the variation for different SZA and RAA. Also shown are the DFS. AVKs stem from retrievals with noisy measurements (v1n).

Optionally, each participant can define criteria for the validity of the retrieved profiles, and can provide corresponding Boolean validity flags for each profile.

## 6.2 Comparison of averaging kernels

Figures 5, 6, 7 and 8 show the AVKs from the OEM algorithms for aerosols at 360 nm and 477 nm, HCHO, and $NO_2$, respectively. The AVKs confirm that MAX-DOAS measurements yield only information on the lowermost $\approx 2$ km of the atmosphere when using a zenith-sky reference spectrum from the same EA sequence. This is a result of the measurement geometry and, in case of aerosols, also of the fact that the $O_4$ vertical distribution is heavily weighted to the surface. The DFS range from $d_s < 0.5$ during foggy conditions (AER8) to $d_s > 3$ for aerosol-free atmospheres (AER0), with the information content being generally higher for aerosols than for trace gases. The dependency of the information content on wavelength is inconsistent: For a pure Rayleigh atmosphere (AER0), bePRO, BOREAS and MMF report smaller DFS for aerosols at 477 nm than at 360 nm, while the opposite is true for HEIPRO and PRIAM. The information content for trace gases increases with wavelength for all algorithms, except for bePRO in case of the AER0 scenario.

The shapes of the AVKs from the different models have a high degree of similarity, except for BOREAS aerosols, where the AVKs show a much smaller information content than all other algorithms and indicate that there is very little height sensitivity for aerosol profiles from the BOREAS algorithm. The respective BOREAS aerosol vertical profiles are, however, in

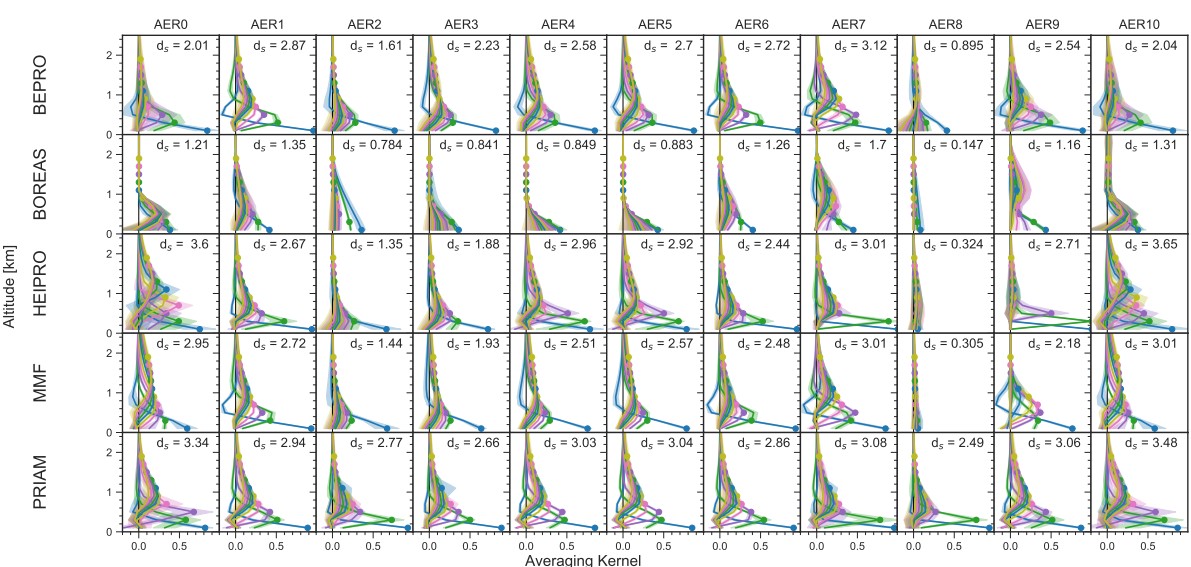

**Figure 6.** Same as Figure 5, but for aerosols at 477 nm.

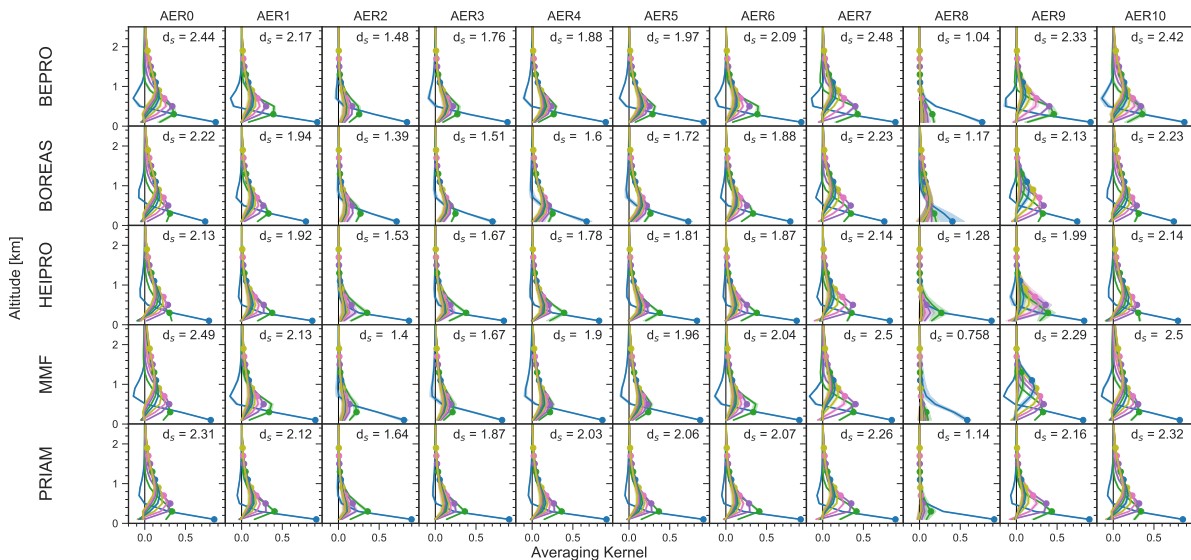

**Figure 7.** Same as Figure 5, but for HCHO.

good agreement with the results from the other algorithms (see Section 6.4). The reason for this apparent discrepancy is that BOREAS is not a standard OEM retrieval but includes additional regularisation terms, making interpretation of the AVKs less straightforward (Bösch et al., 2018). Significant differences between the AVKs can be found for the rather extreme scenarios

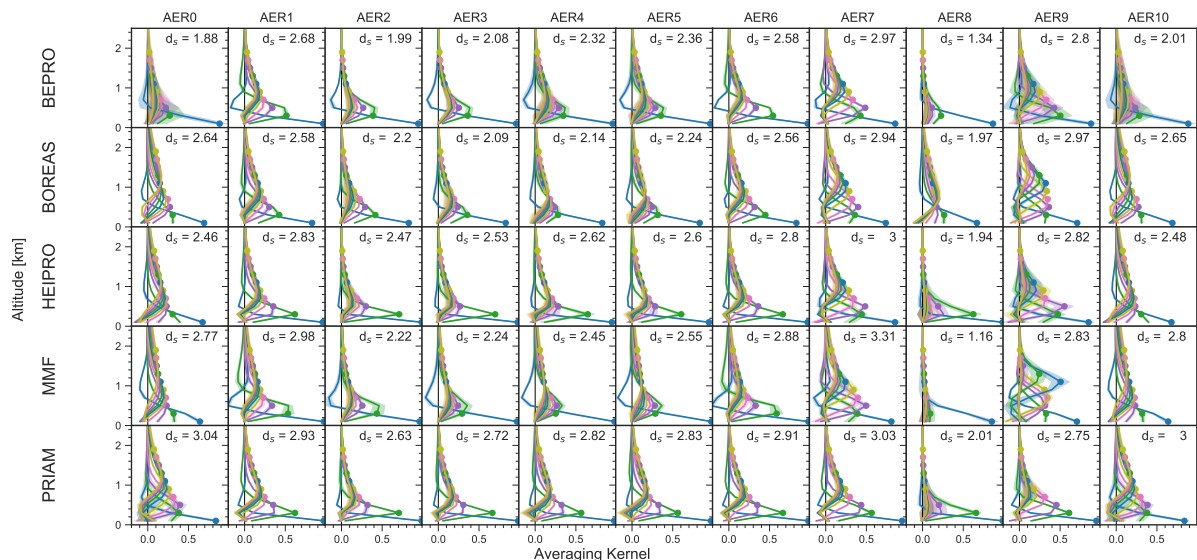

**Figure 8.** Same as Figure 5, but for $NO_2$.

AER8 and AER9 (fog and low-lying cloud). PRIAM aerosol AVKs are much less affected by fog (AER8) than the other algorithms, which show a strong reduction in information content in case of high extinction at the surface. The AVKs for a high-lying cloud at $\approx 5$ km altitude (AER10) are very similar to those of a Rayleigh atmosphere (AER0), indicating that horizontally homogeneous free tropospheric clouds have little impact on the sensitivity of MAX-DOAS retrievals.

Apart from the fog scenario (AER8), there is only a moderate dependency of vertical resolution and information content on the aerosol content of the atmosphere. Interestingly, the information content for trace gases, in particular $NO_2$, is highest for the uplifted aerosol layer (AER7). This layer probably increases the sensitivity for trace gases near the surface due to the fact that the majority of scattering occurs within the cloud. The light paths below the cloud are thus well defined and, in case of low EAs, can be higher than for the clear-sky case.

The variability of the AVKs with the position of the Sun (shown as shaded areas in Figures 6 - 8) is generally very small, indicating little dependency on SZA and RAA.

### 6.3 Comparison of a posteriori dSCDs

An important indicator for the level of convergence of the retrieval, and subsequently the accuracy of the retrieved profile, is the agreement between the measurement vector $\mathbf{y}$ and the measurement vector $\mathbf{F}(\hat{\mathbf{x}})$ modelled for the retrieved state $\hat{\mathbf{x}}$ (a posteriori
dSCDs). The comparison of a posteriori and measured dSCDs for the v1n dataset is shown in Figure 9, and the corresponding linear regression parameters are depicted in Figure 10. The NASA algorithm does not rely on the forward modelling of dSCDs, thus no dSCD data is available for this algorithm.

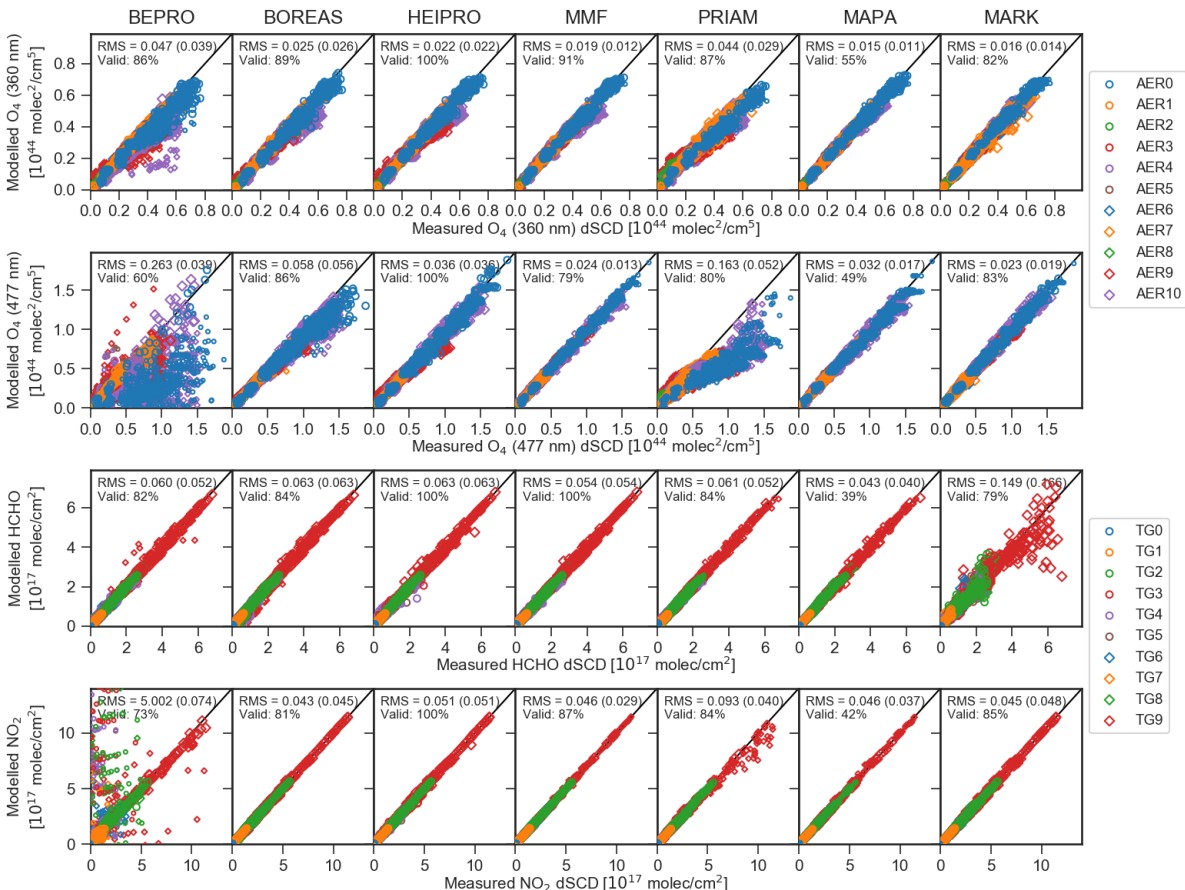

**Figure 9.** A posteriori versus measured dSCDs of $O_4$ (360 nm and 477 nm), HCHO and $NO_2$ for the v1n dataset. Colours and symbols indicate the underlying aerosol and trace gas profiles, respectively, as denoted in the legend. Data flagged as invalid is shown as smaller symbols. Also shown are the RMS difference between measured and modelled dSCDs with the RMS for valid data only being shown in brackets, as well as the percentage of valid data. Units of the RMS are $10^{44}$ molec$^2$/cm$^5$ for $O_4$ and $10^{17}$ molec/cm$^2$ for HCHO and $NO_2$.

The algorithms show significant differences in the level of convergence. BOREAS, HEIPRO, MMF and MAPA show good agreement between measured and modelled dSCDs for all scenarios with slopes and Pearson correlation coefficients close to unity. The same holds true for MARK except for the HCHO retrieval, where poor convergence is achieved for the TG9 scenario, but this has only little effect on the regression parameters (see Figure 10). PRIAM achieves good convergence for the trace gas retrievals, but shows a larger scatter than other algorithms for $O_4$ at 360 nm and significantly underestimates $O_4$ at 477 nm, where the slope of the linear regression is only 0.585. In many cases, bePRO yields only poor agreement between measured and modelled dSCDs, in particular in the visible where the RMS between measurement and modelling is significantly higher than for the other algorithms, and regression coefficients are only 0.67 for $O_4$ dSCDs and 0.24 for $NO_2$ dSCDs. bePRO furthermore has problems retrieving the AER10 scenario (high-lying cloud) at 360 nm. The poor convergence of the bePRO algorithm is

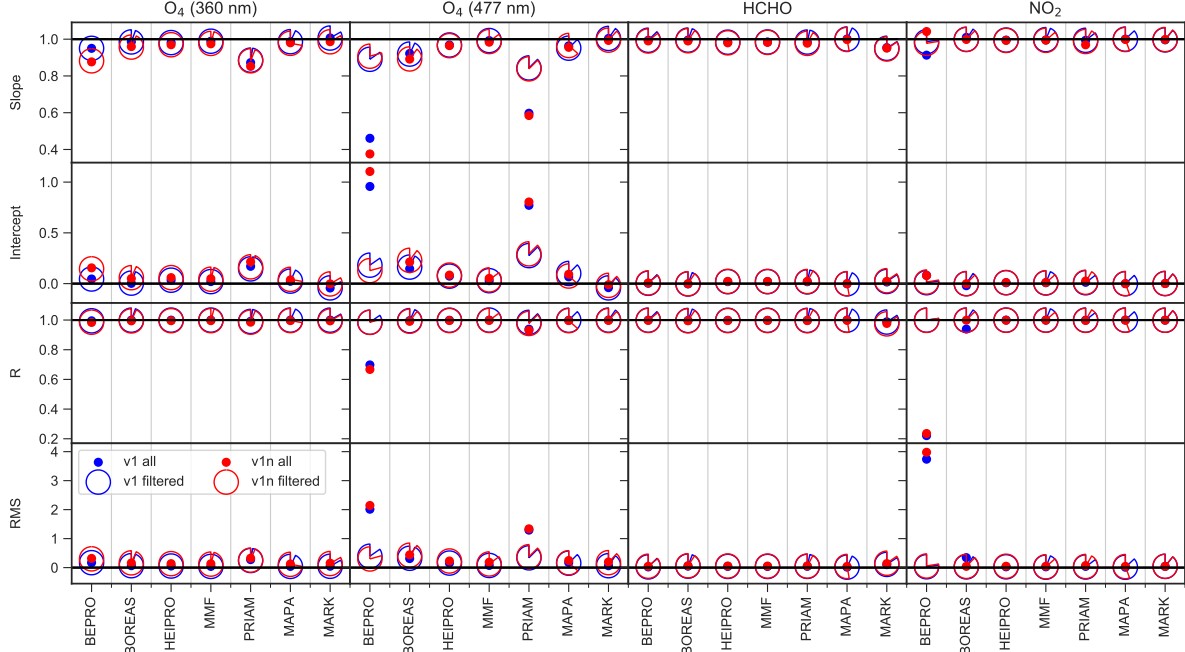

**Figure 10.** Slope, intercept, regression coefficient and RMS of the dSCD correlation. Each of the circular symbols for the filtered data represents a pie chart that quantifies the fraction of data flagged as valid. RMS and intercept values are in units of $10^{43}$ molec$^2$/cm$^5$ for $O_4$, and $10^{17}$ molec/cm$^2$ for HCHO and $NO_2$. Cloud and fog scenarios (AER8, AER9, AER10) are excluded from the regression analysis.

the result of numerous outliers, and the RMS strongly improves after discarding about 15% of the 477 nm $O_4$ datapoints and 23% of the $NO_2$ datapoints that are flagged as invalid. The reason for the instabilities of the bePRO algorithm, which strongly affect the accuracy of the retrieved profiles (see Section 6.4), is the fact that bePRO uses a Gauss-Newton inversion scheme. Sensitivity studies have shown that this problem could be overcome by using a Levenberg-Marquardt inversion scheme instead.

5    More than 55% of data from MAPA is flagged as invalid although the retrievals show a high level of convergence. This might indicate that the MAPA quality flagging criteria could be too strict.

### 6.4 Comparison of vertical profiles

In this section, the overall ability of the retrieval algorithms to reproduce the true atmospheric aerosol and trace gas profiles is discussed. Figures 11 and 12 show the comparison between true and retrieved aerosol profiles at 360 nm and 477 nm,

10    and HCHO and $NO_2$ profile comparisons are shown in Figures 13 and 14, respectively. The regression parameters for the comparison of true and retrieved aerosol extinction and trace gas number concentration for all target species are presented in Figure 15. All data presented here are based on retrievals with noisy dSCDs (v1n). The corresponding profiles with valid data only, and with noise-free measurements, are shown in the supplemental material.

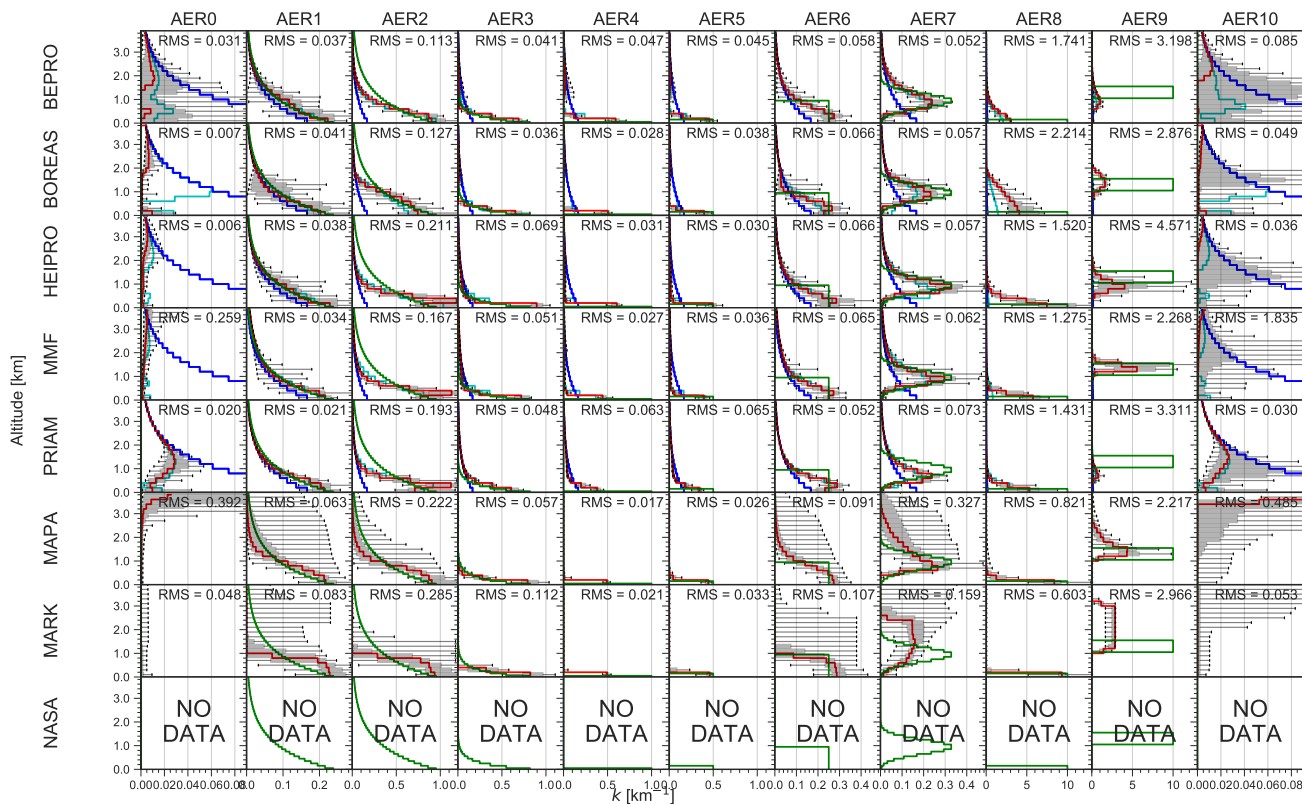

**Figure 11.** Comparison of retrieved (red solid line, version v1n) and true (green solid line) vertical profiles of aerosol extinction at 360 nm for each aerosol scenario (columns) and algorithm (rows). The retrieved profiles are the medians for all SZA/RAA combinations. The (25% - 75%) and (5% - 95%) percentiles are shown as gray areas and whiskers, respectively. The a priori profile is shown as blue line (OEM algorithms only).

Most algorithms are capable of realistically retrieving the shape of the true aerosol extinction profiles for moderate conditions (AER0 .. AER7), with slopes close to unity and Pearson regression coefficients for the correlation between true and retrieved extinction of $R > 0.8$ (see Figure 15). Exponential profiles (AER1, AER2, and AER3) are well reproduced, but the extinction above $\approx 1$ km is underestimated by all algorithms for AER2 (high aerosols, exponentially decreasing) owing to a reduced sensitivity of MAX-DOAS measurements to high altitudes. In the case of OEM algorithms, this leads to a bias towards the a priori as discussed in Section 6.2, whereas parametrised algorithms tend to become unstable at altitudes where the sensitivity is low. The shallow aerosol layers AER4 and AER5 are captured well by all algorithms except NASA (which provides profiles at 477 nm only). The retrievals are performed on a 200 m grid, and the 100 m thick layer (AER4) yields results similar to the 200 m thick layer (AER5), which has the same AOT but only half the extinction coefficient. As expected, the 1 km thick aerosol layer AER6 is significantly smoothed by the retrievals, except by MARK, which reproduces the sharp edges of this profile

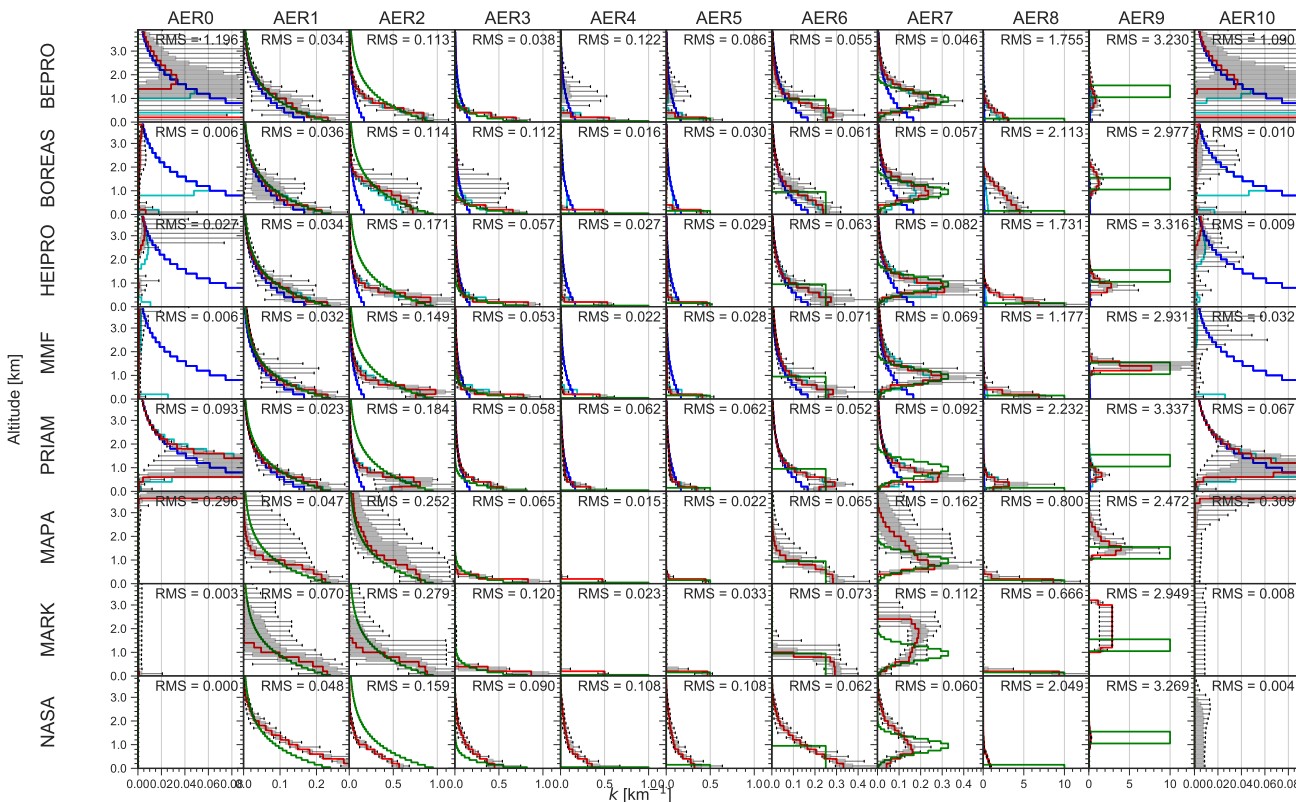

**Figure 12.** Same as Figure 11, but for aerosols at 477 nm.

better than the other algorithms. The uplifted Gaussian profile with a centre altitude of 1 km (AER7) is well reproduced by all OEM algorithms, and to a lesser extent also by NASA. The parametrised algorithms MAPA and MARK also retrieve uplifted profiles, but significantly overestimate the extinction above 1 km.

The 200 m thick fog layer (AER8) is very well reproduced by the parametrised algorithms MAPA and MARK, and to a

5 lesser extent also by the OEM algorithms MMF, HEIPRO and PRIAM, which retrieve the logarithm of the aerosol extinction profile and are therefore capable of retrieving a higher range of aerosol extinction values than bePRO, which operates in linear space and therefore are subject to a stronger bias towards the a priori. The cloud above 1 km (AER9) is very well retrieved by MMF. MAPA and MARK capture the cloud bottom altitude but overestimate the extinction above the cloud. bePRO, BOREAS, HEIPRO and PRIAM retrieve only a small enhancement of the aerosol extinction ($< 2\,\mathrm{km}^{-1}$) around the altitude of the cloud,

10 and NASA appears to be insensitive to the cloud.

The retrieved extinction profiles for scenario AER10, which consists of a cloud above 5 km altitude and no aerosols elsewhere, are very similar to the cloud- and aerosol-free scenario AER0, although the RMS difference between true and retrieved profiles are higher for AER10 than for AER0 in some cases. As already discussed in Section 6.3, measured and a posteriori

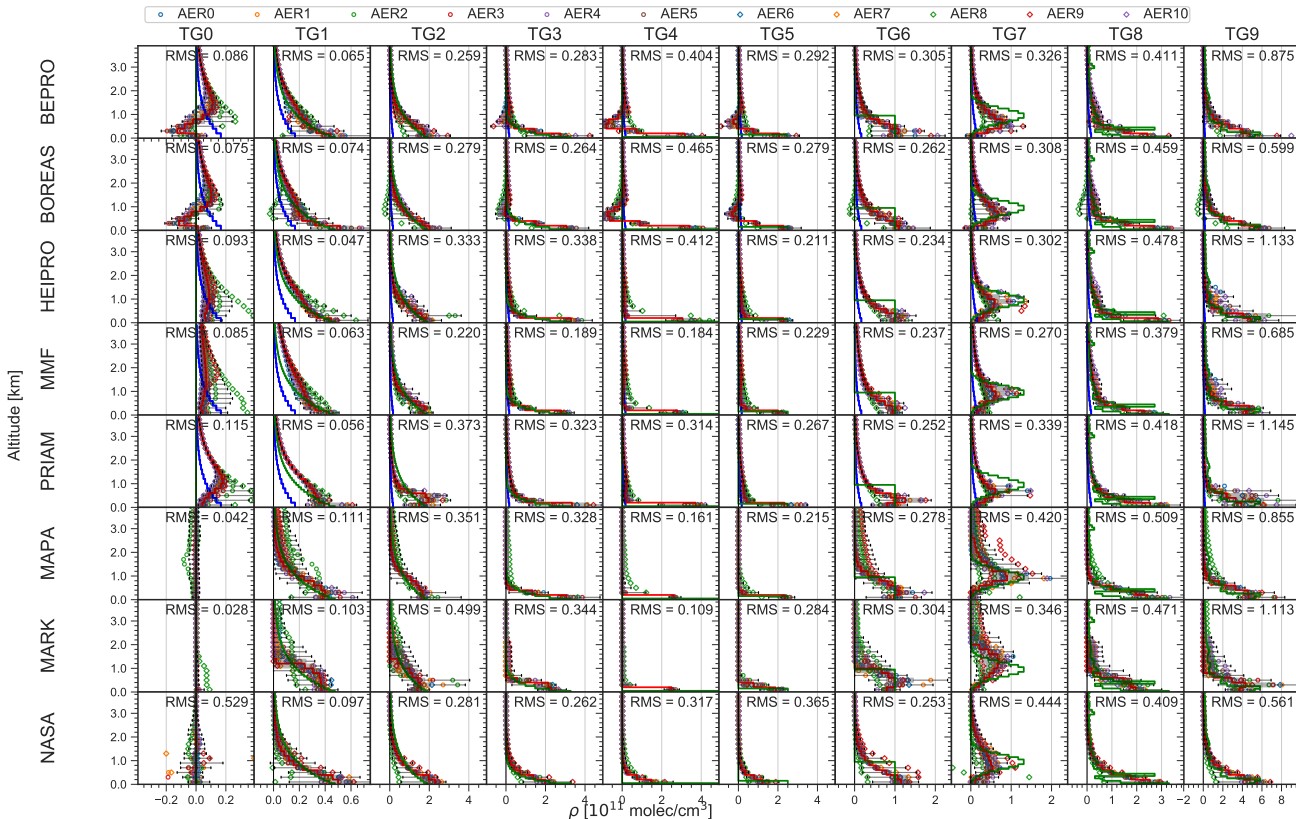

**Figure 13.** Comparison of retrieved (red solid line, version v1n) and true (green solid line) vertical profiles of HCHO for each trace gas scenario (columns) and algorithm (rows). In addition to the median profiles for all aerosol scenarios, SZAs and SAAs shown in red, the median concentration profile for each aerosol scenario is shown as coloured symbols as denoted in the legend. The RMS (true - retrieved extinction) is shown in units of $10^{11}$ molec/cm$^3$.

dSCDs are also in good agreement for AER10. It can therefore be concluded that high-lying clouds have very little effect on MAX-DOAS aerosol retrievals, in contradiction to the findings of Ortega et al. (2016), who suggest that free-tropospheric aerosols and clouds would strongly affect MAX-DOAS measurements of $O_4$.

As can be seen from the width of the 50% and 90% confidence intervals (shaded areas and error bars in Figures 11 and 12),
5   the parametrised algorithms (MAPA and MARK) produce a significant number of outliers of the retrieved aerosol profiles, in particular for the scenarios AER1, AER2, AER6 and AER7. This effect is larger for retrievals with noisy than with noise-free dSCDs (see supplemental Figures S5 and S6). A likely reason for this behaviour is that MAPA and MARK do not rely on a priori constraints as a regularisation, but use a minimisation method for the differences between measured and modelled dSCDs. In some cases the ensemble of possible solutions with similar minima can be wide-spread. However, in most cases

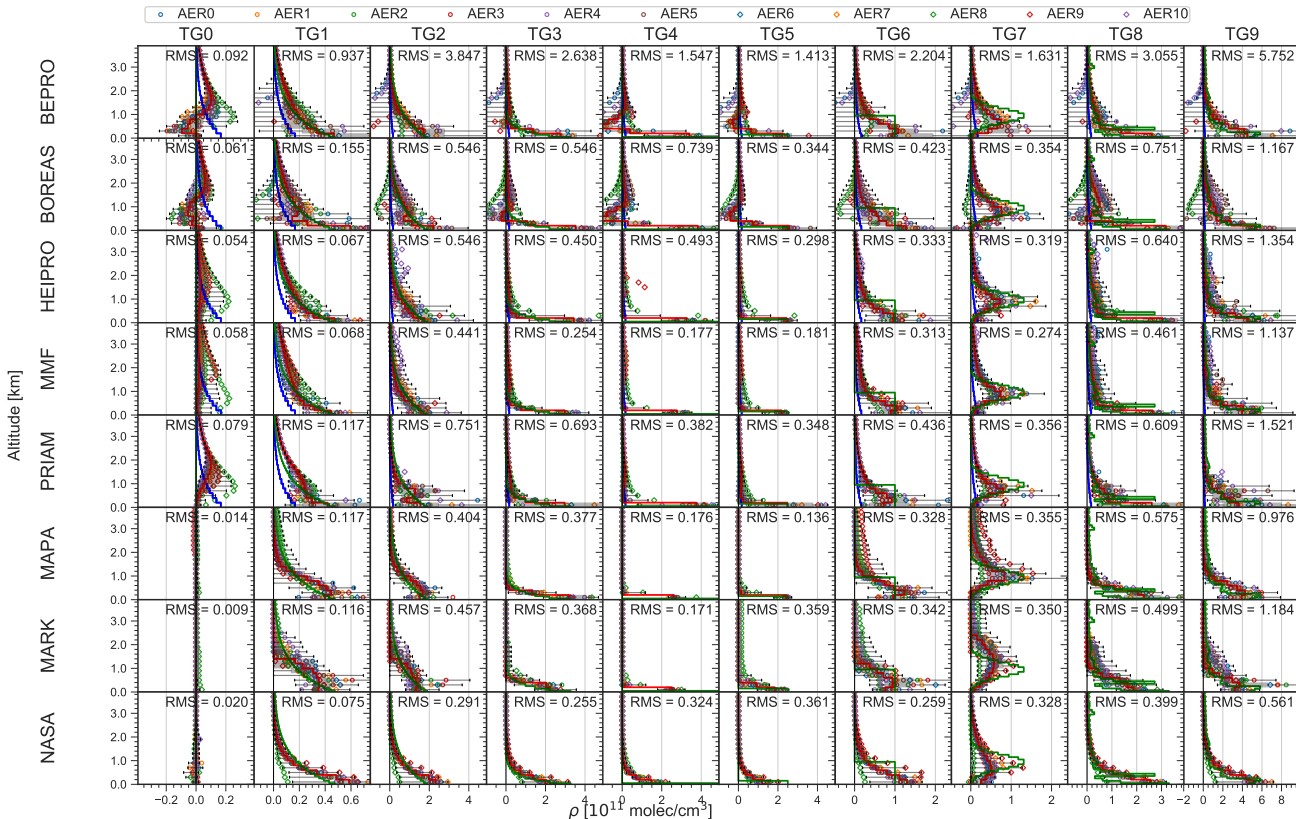

**Figure 14.** Same as Figure 13, but for $NO_2$.

the outliers are not significantly affecting the overall accuracy as expressed by the RMS difference between retrieved and true aerosol profiles (see Figure 15).

Note that for MAPA the spread of the retrieved profiles is far smaller when if only filtered results are considered (see supplemental Figures S1 - S4), indicating that the MAPA filter is successfully removing outliers. This requires, however, excluding 17% and 37% of the aerosol profiles at 360 nm and 477 nm, respectively, as well as 47% of the HCHO profiles and 44% of the $NO_2$ profiles of the AER0 - AER7 scenarios, while almost 93% of the profiles for scenarios with cloud and fog layers (AER8 - AER10) are discarded.

As a result of a poor convergence of modelled and measured dSCDs (see Section 6.3), bePRO fails to retrieve the extinction profile at 477 nm for aerosol-free atmospheres (AER0) and for a high-lying cloud (AER10). The RMS difference between true and retrieved extinction is significantly higher for bePRO than for other algorithms for AER4 (100 m thick aerosol layer). As a result of these instabilities, the linear regression of retrieved and true aerosol extinction coefficients at 477 nm yields smaller slope (0.76) and regression coefficient (0.26) for bePRO than for the other algorithms when including all data. After filtering out $\approx 75\%$ of data flagged as invalid, the accuracy of the bePRO results is similar to the other algorithms.

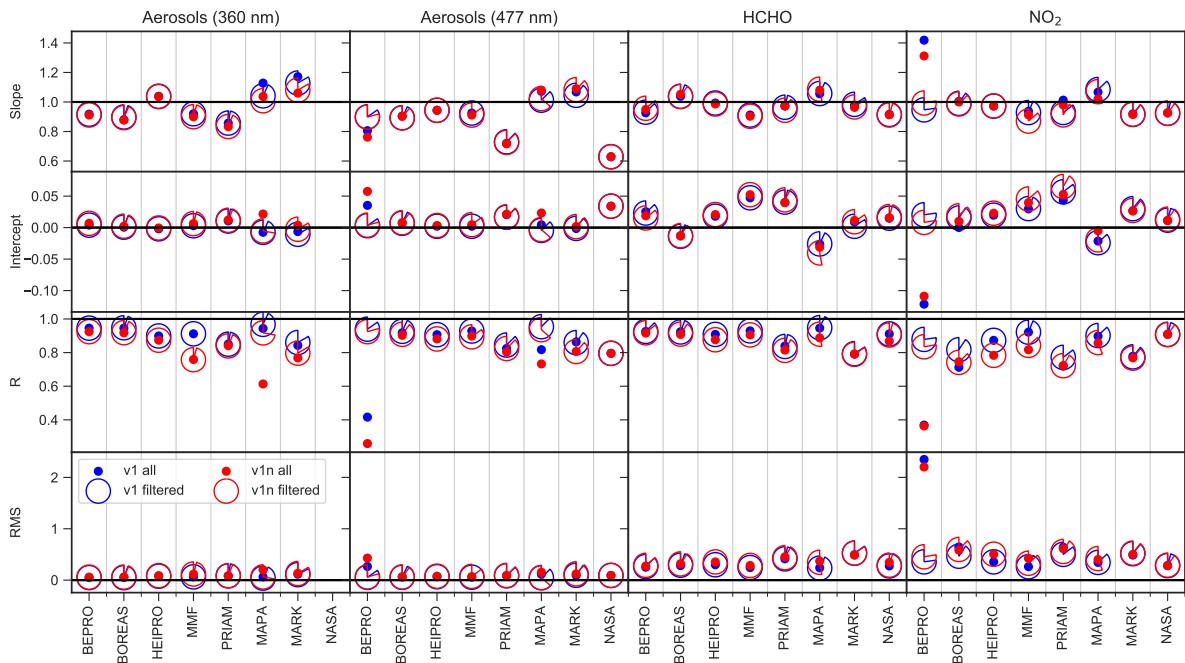

**Figure 15.** Slope, intercept, regression coefficient and RMS for the correlation between true and retrieved aerosol extinction as well as HCHO and $NO_2$ number concentrations at all altitudes. Intercept and RMS values are in units of $km^{-1}$ for aerosols and $10^{11}$ molec/cm$^3$ for HCHO and $NO_2$. Cloud and fog scenarios (AER8, AER9, AER10) are excluded from the regression analysis.

PRIAM and NASA underestimate the aerosol extinction at 477 nm with slopes of only 0.72 and 0.63, respectively. Furthermore, PRIAM falsely retrieves a non-existing uplifted aerosol layer around 1 km altitude with a peak extinction $> 0.1$ km$^{-1}$ for aerosol-free atmospheres (AER0 and AER10) at 477 nm, whereas MAPA retrieves elevated aerosol extinction at the top of the retrieval domain between 3.5 km and 4 km for these scenarios both at 360 nm and 477 nm. This is a consequence of the

fact that there are no a priori constraints on extinction profiles made in MAPA, and the resulting dSCDs are almost unaffected by high cloud layers. Most of the MAPA profiles for the AER10 scenario are, however, flagged as invalid, leaving only 2% of valid profiles (see supplemental Figures S1 and S2).

The retrieval is generally more stable for trace gases (Figures 13 and 14) than for aerosols. Obviously, it is sufficient to constrain the trace gas retrieval with an aerosol profile that realistically reproduces the light path, even if the extinction profile

differs from the truth. Furthermore, trace gas retrievals are probably more stable because they constitute a linear problem in contrast to the non-linear aerosol retrieval.

With the exception of $NO_2$ from bePRO, the algorithms reproduce exponential trace gas profiles (TG1, TG2 and TG3) well, with little bias towards the a priori at high altitudes (Figures 13 and 14). As for aerosols at 477 nm, bePRO $NO_2$ profiles suffer from significant instabilities, and BOREAS shows oscillating features with negative values for altitudes between 1

and 2 km. These discrepancies of the bePRO and BOREAS results are probably due to the fact that they operate in linear

space. The vertical extent and trace gas concentration of the shallow trace gas layers TG4 and TG5 are well reproduced by all algorithms except NASA. HEIPRO $NO_2$ results for TG4 suffer from outliers for the AER9 scenario, where $NO_2$ above the cloud is overestimated. As for aerosols, the degree of smoothing of the 1 km box profile (TG6) is higher for the OEM than for the parametrised algorithms. The peak altitude of the uplifted profile (TG7) at 360 nm is underestimated by all algorithms

except MAPA and MARK, with the latter overestimating the altitude of the trace gas layer. The agreement between retrieved and true TG7 profiles is better for $NO_2$, where HEIPRO, MMF and MAPA retrieve the peak altitude at the right location, than for HCHO. This is probably owing to the lower information content and lower sensitivity for high altitudes in the UV compared to the visible (see Section 6.2). The uplifted trace gas profile (TG7) is subject to a larger degree of smoothing than the corresponding uplifted aerosol profile (AER7). Owing to the limited vertical resolution of MAX-DOAS measurements,

the fine structure of the trace gas profiles measured by a $NO_2$ balloon sonde (TG8 and TG9) is not well reproduced by the retrievals. The lowest level of agreement between true and retrieved trace gas profiles is found for the fog scenario (AER8), since the high extinction at the ground leads to a very small information content and a low sensitivity for trace gases except for the lowermost atmospheric layers (see Section 6.2).

MAPA, MARK and NASA algorithms retrieve profiles close to zero for the trace-gas free atmospheres (TG0), whereas the

OEM algorithms either exhibit a slight bias towards the a priori (HEIPRO, MMF, PRIAM), or oscillate around zero (bePRO, BOREAS) for this scenario. Sensitivity studies based on the MMF algorithm have shown that these oscillations are suppressed if the logarithm of the profile is retrieved, as is the case for HEIPRO, MMF and PRIAM. This representation also prevents the retrieval of negative values, which occur for bePRO and BOREAS, in particular for TG0 and TG4.

Except for bePRO $NO_2$, a linear regression between true and retrieved trace gas concentrations yields slopes between 0.9 and

1.08 (see Figure 15). Regression coefficients usually exceed 0.8, except for $NO_2$ in the case of bePRO ($R = 0.36$), BOREAS ($R = 0.75$), PRIAM ($R = 0.72$) and MARK ($R = 0.77$). In particular at 360 nm, MMF and MARK aerosols appear to be more sensitive to noise than the other algorithms, with Pearson's R decreasing from 0.93 to 0.76 (MMF) and from 0.84 to 0.77 (MARK) after adding noise to the synthetic measurements. In general, $NO_2$ profiles from the OEM algorithms appear to be more sensitive to measurement noise than HCHO profiles.

**6.5    Comparison of total columns**

In this section, the ability of the retrieval algorithms to retrieve aerosol and trace gas total columns is discussed. Box-whisker plots comparing retrieved and true AOT at 360 nm and 477 nm are shown in Figure 16. The corresponding plots for HCHO and $NO_2$ VCDs are shown in Figure 17. The parameters of a linear regression between retrieved and true total columns are shown in Figure 18. Note that the total column is defined here as the integral of extinction coefficient and number concentration

within the retrieval domain, i.e. from the surface to 4 km altitude. This means that, e.g., a high-altitude cloud above the retrieval domain as in the AER10 scenario is not considered in the calculation of the AOT.

The total column of both trace gases and aerosols is retrieved accurately by most algorithms. Except for foggy conditions (AER8), there is little dependency of the accuracy of the retrieved trace gas VCD on the aerosol profile. As expected from the limited sensitivity to high altitudes, the total columns from OEM algorithms tend to be biased towards the a priori. For the

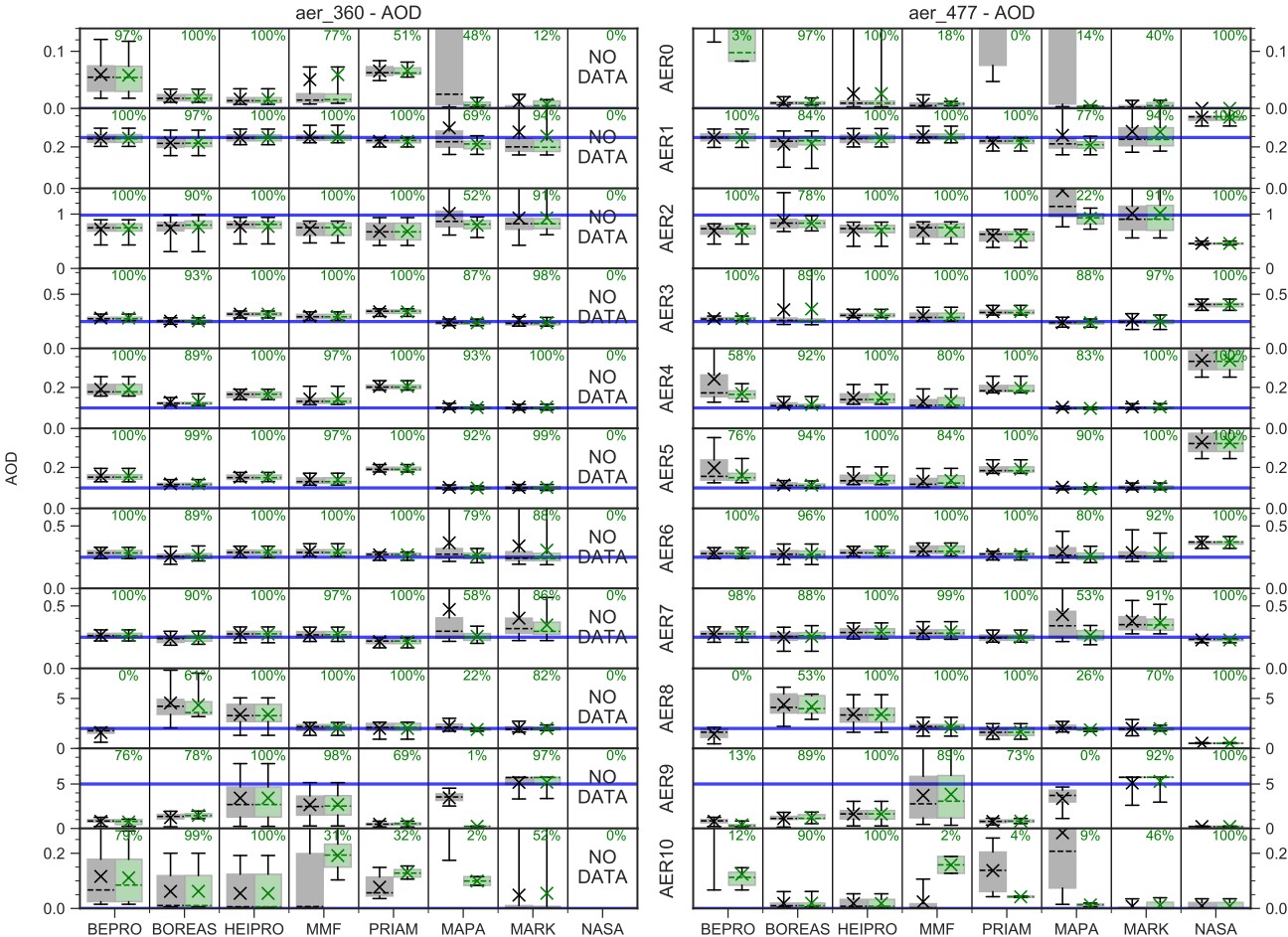

**Figure 16.** Box-whisker plots of the retrieved AOT at 360 nm (left) and 477 nm (right) for the different aerosol scenarios (rows) and retrieval algorithms (columns) based on retrievals using noisy dSCDs. Crosses show the mean, dashed horizontal lines the median. Shaded areas indicate the (25% -75%) percentile, whiskers the (5% - 95%) percentile. The true AOT is shown as blue horizontal line. For each algorithm, all data are shown on the left in black, and data marked as valid on the right in green. Also shown is the percentage of valid data.

aerosol and trace gas free scenarios AER0 and TG0, a positive bias of (0.02 - 0.06) for AOT and $(0.2 - 0.35) \cdot 10^{16}$ molec/cm$^2$ for trace gases is found for the OEM algorithms. Ranging from 0.05 to 0.15, the AOT bias is somewhat higher for AER10 with a cloud above 5 km altitude than for AER0. The OEM algorithms show a negative bias for the high extinction scenario AER2 (AOT of 1), and a positive bias for the shallow layers AER4, AER5 and TG4, TG5. The bias towards the a priori is

5    furthermore reflected by a smaller slope of the linear regression between retrieved and true total column for the OEM algorithms compared to the parametrised and analytical algorithms MAPA, MARK and NASA for all species except NO$_2$ (see Figure 18). Again, bePRO shows poor performance in the visible with regression coefficients of 0.02 and 0.71 for aerosols at 477 nm and

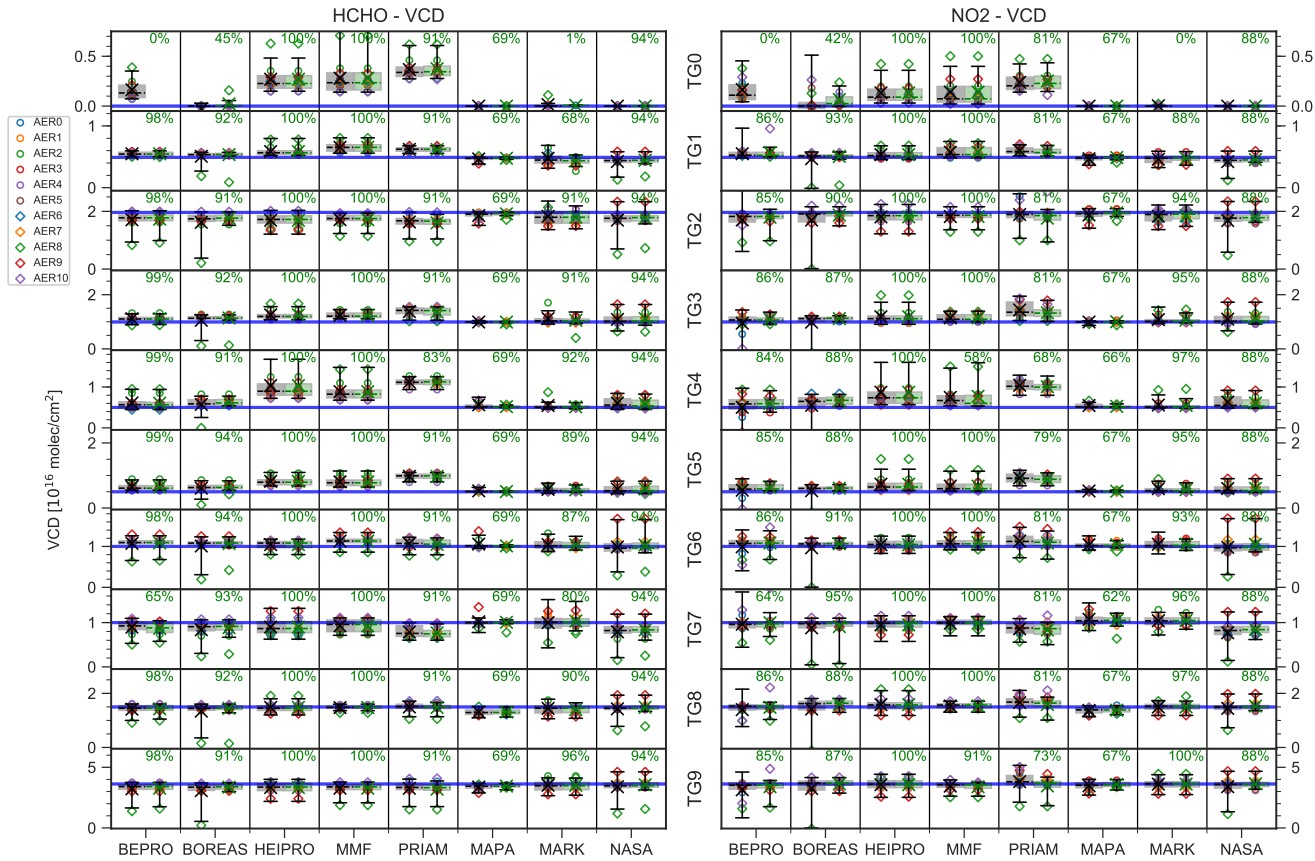

**Figure 17.** Box-whisker plots of the retrieved HCHO (left) and NO$_2$ (right) VCD for the different trace gas scenarios (rows) and retrieval algorithms (columns) based on retrievals using noisy dSCDs. Crosses show the mean, dashed horizontal lines the median. Shaded areas indicate the (25% -75%) percentile, whiskers the (5% - 95%) percentile. The true VCD is shown as blue vertical line. For each algorithm, all data is shown on the left in black, and data marked as valid on the right in green. Also shown is the percentage of valid data. The median for each aerosol scenario is shown as coloured symbol as indicated in the legend.

NO$_2$, respectively, while all other algorithms yield $R > 0.85$ and $R > 0.93$ for aerosols and trace gases, respectively. These discrepancies mainly occur for the AER0 scenario, and to a lesser extent also for AER4.

The parametrised algorithms MAPA and MARK accurately retrieve the total column in most cases. Exceptions are the AER6 and AER7 scenarios (1 km box profile and uplifted profile), where both algorithms show a positive bias. Both MAPA and MARK show a significant scatter of the AOT for AER0, AER1, AER2, AER7 and AER10 that, in case of MAPA, is reduced by filtering out a significant fraction of the data (28% at 360 nm and 37% at 477 nm). Furthermore, MAPA significantly overestimates the AOT at 477 nm with a slope of 1.4, and yields a regression coefficient of only 0.56 for the AOT at 360 nm when considering all data. Regression parameters for MAPA are, however, comparable to those from the other algorithms

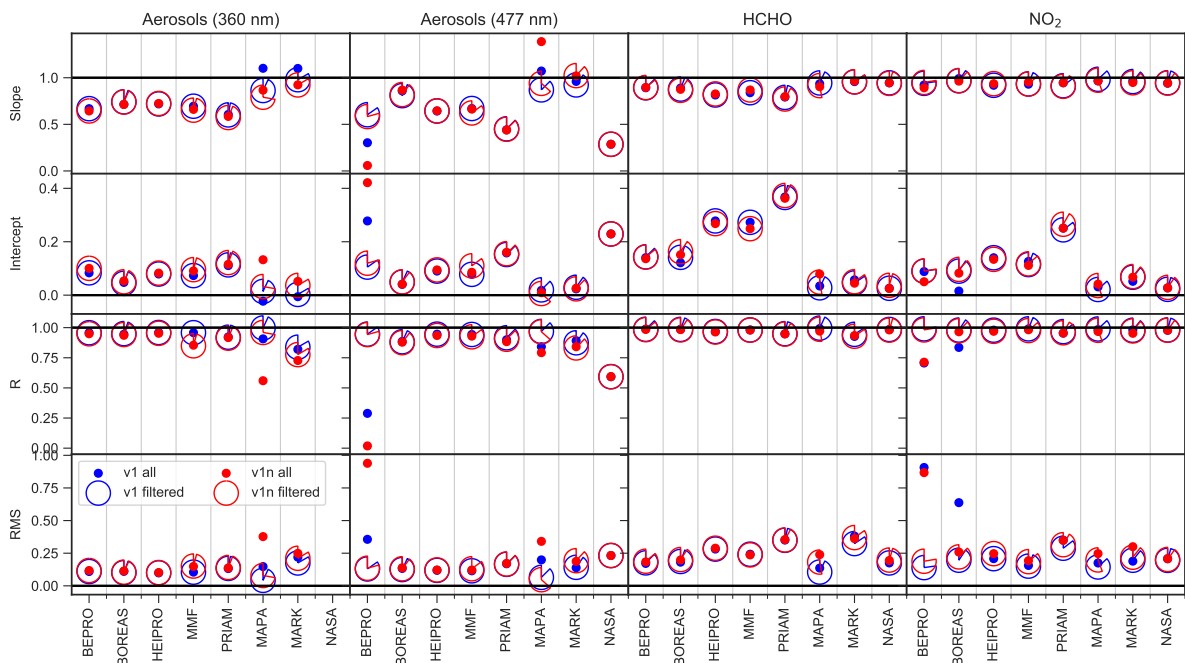

**Figure 18.** Slope, intercept, regression coefficient and RMS for the correlation between true and retrieved AOT as well as HCHO and $NO_2$ VCD. Intercept and RMS values are in dimensionless units for aerosols and $10^{16}$ molec/cm$^2$ for HCHO and $NO_2$. Cloud and fog scenarios (AER8, AER9, AER10) are excluded from the regression analysis.

after removing data flagged as invalid. MARK flags almost all TG0 data as invalid, although the agreement between true and retrieved VCD is very good for this algorithm. The NASA algorithm retrieves the HCHO and $NO_2$ VCD accurately, but shows a significant negative bias for all aerosol scenarios at 477 nm (except for AER6), where the regression coefficient is less than 0.6.

## 6.6 Comparison of surface values

In this section, the agreement between true and retrieved surface extinction and surface concentration (i.e., the values in the lowermost layer of the respective profiles with a thickness of 200 m) are discussed. As for the total column discussed in the previous section, Figures 19 and 20 show box-whisker plots for the aerosol and trace gas surface values, and Figure 21 lists the parameters of the linear regression between true and retrieved aerosol extinction and trace gas concentration in the lowermost retrieval layer.

Surface aerosol extinction and trace gas profiles are generally well reproduced. For bePRO retrievals in the visible, however, a negative slope and no significant correlation ($R = -0.098$) between true and retrieved values are found for surface aerosol extinction at 477 nm. Furthermore, the regression coefficient for bePRO $NO_2$ is only 0.24 due to the aforementioned deviations in case of the AER0 and AER4 scenarios. However, the regression parameters of the bePRO algorithm significantly improve

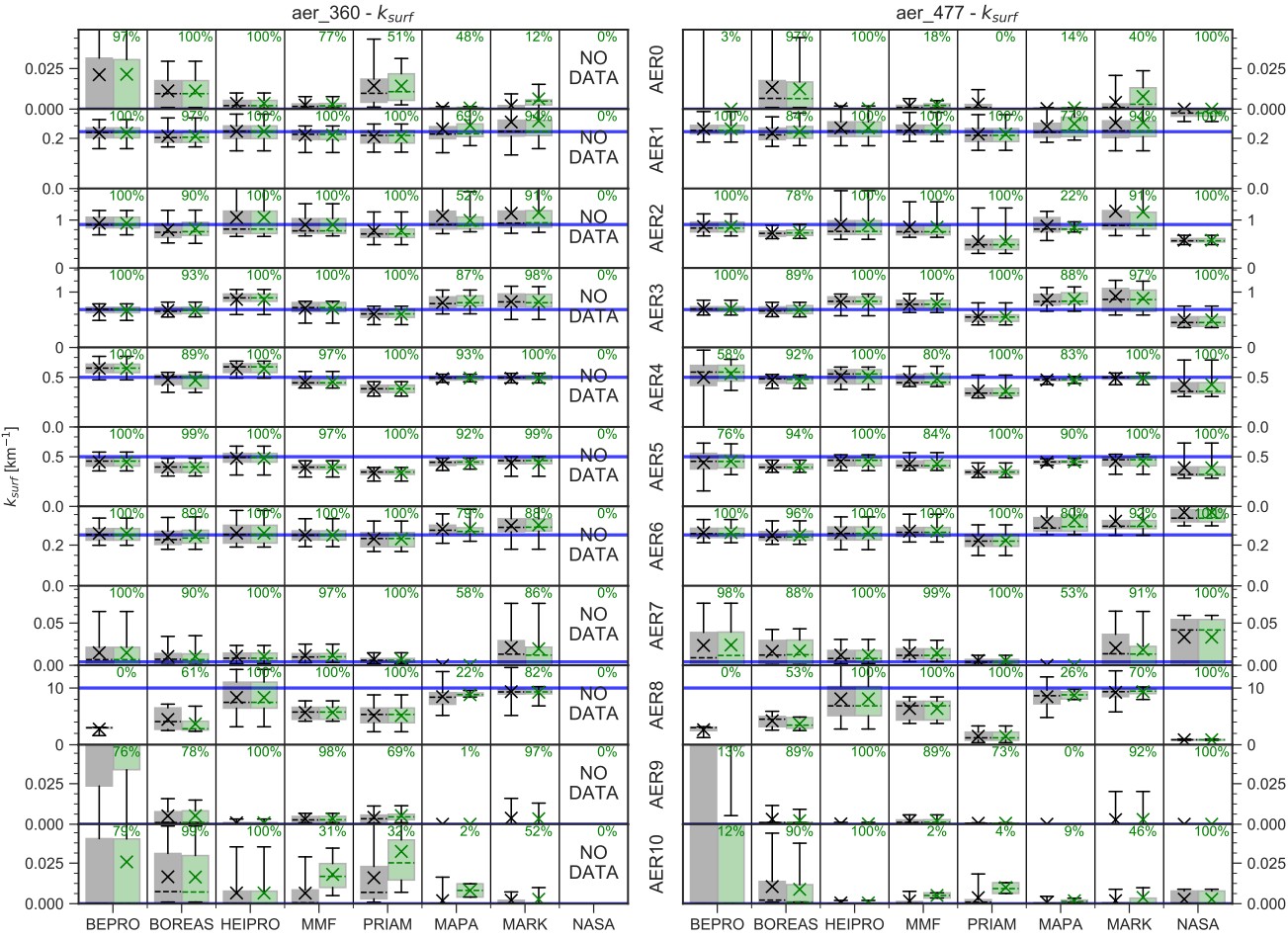

**Figure 19.** Box-whisker plots of the retrieved surface aerosol extinction at 360 nm (left) and 477 nm (right) for the different aerosol scenarios (rows) and retrieval algorithms (columns) based on retrievals using noisy dSCDs. Crosses show the mean, dashed horizontal lines the median. Shaded areas indicate the (25% -75%) percentile, whiskers the (5% - 95%) percentile. The true AOT is shown as blue vertical line. For each algorithm, all data are shown on the left in black, and data marked as valid on the right in green. Also shown is the percentage of valid data.

after discarding about 20% of the profiles marked as invalid. The OEM algorithms yield a small positive bias for scenarios with no or only small surface extinction coefficients and trace gas concentrations (AER0, AER7, AER9, AER10, TG0 and TG7). PRIAM and NASA both underestimate the aerosol extinction at 477 nm (slope between 0.6 and 0.65, respectively). In contrast, HEIPRO, MAPA and MARK overestimate the surface extinction, in particular at 360 nm (slope $> 1.16$), and yield 5 smaller regression coefficients ($R < 0.9$ at 360 nm) than the other algorithms.

For the prescribed profile scenarios, retrieved aerosol surface extinction (mean regression coefficient from all algorithms $\bar{R} = 0.903$ at 360 nm and $\bar{R} = 0.825$ at 477 nm ) shows a better agreement with the true value that the AOT ($\bar{R} = 0.874$ at 360

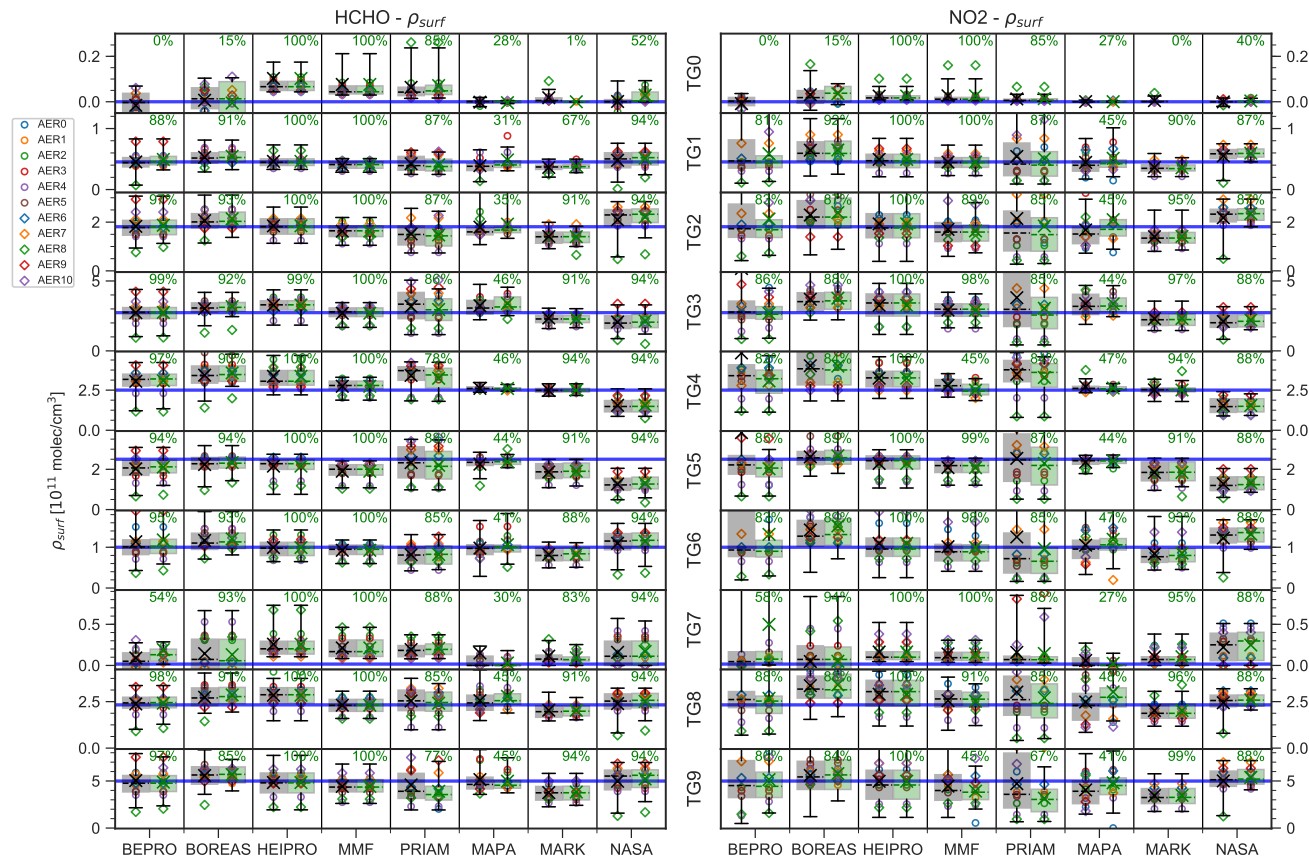

**Figure 20.** Same as Figure 19, but for HCHO (left) and NO₂ (right) surface concentrations. The median for each aerosol scenario is shown as coloured symbol as indicated in the legend.

nm and $\bar{R} = 0.803$ at 477 nm). The opposite is true for trace gases, where the agreement of the total column ($\bar{R} = 0.969$ and $\bar{R} = 0.955$ for HCHO and NO₂, respectively) is better than for the surface mixing ratio ($\bar{R} = 0.894$ and $\bar{R} = 0.780$).

## 6.7 Computational performance

In order to assess the numerical performance of the different retrieval algorithms, the duration for a single profile retrieval was reported by each participant. For multi-processor systems, the total time has been multiplied by the number of processor kernels used for the retrieval. It is important to note that the retrievals were performed individually by each participant, using computers with different performances. A more accurate comparison would require to run all algorithms on the same computer which is outside the scope of this study. The results of the benchmark test are shown in Figure 22.

The OEM algorithms, which rely on on-line radiative transfer calculations, require between 4 s (MMF) and 23 s (PRIAM) for the retrieval of a single trace gas profile. The duration for the retrieval of an aerosol profile ranges between 6 s for MMF and

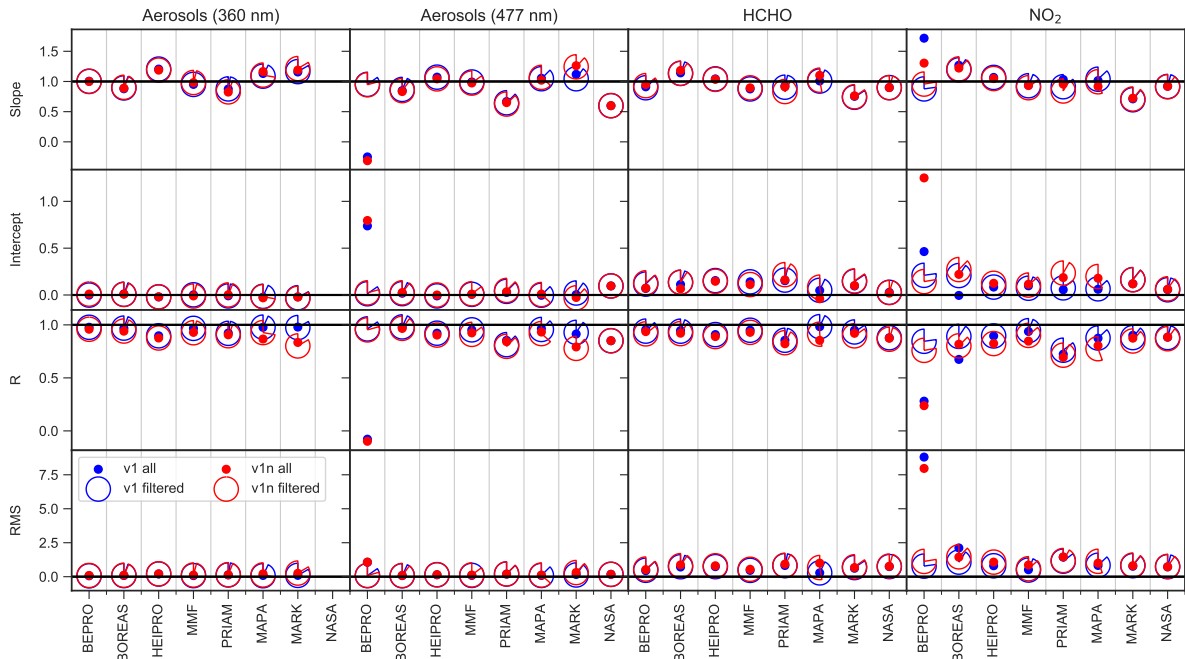

**Figure 21.** Slope, intercept, regression coefficient and RMS for the correlation between true and retrieved aerosol extinction as well as HCHO and NO$_2$ surface concentration. Intercept and RMS values are in dimensionless units for aerosols and $10^{11}$ molec/cm$^3$ for HCHO and NO$_2$. Cloud and fog scenarios (AER8, AER9, AER10) are excluded from the regression analysis.

more than 3.5 minutes for BOREAS. The large range of computational effort for aerosols probably results from the different approaches for the calculation of the weighting functions. The BOREAS aerosol retrieval relies on radiative transfer simulations at several wavelengths, resulting in lowest computational performance, followed by HEIPRO, whose aerosol weighting function calculation is based on the finite difference method, leading to about one minute for an aerosol retrieval, while MMF

5 and bePRO are significantly faster since they rely on analytically calculated aerosol weighting functions.

The parametrised algorithms MAPA and MARK show significant differences in computational performance, although both rely on LUTs for the weighting functions. MARK requires 13 s and 24 s for aerosol and trace gas retrievals, respectively, while MAPA aerosol and trace gas retrievals are executed within 3 s and 2 s, respectively.

The NASA algorithm, which does not rely on radiative transfer modelling but on an analytical approach, is outstanding in

10 terms of computational performance. The retrieval of a single aerosol or trace gas profile requires less than 5 milliseconds, and is thus almost three orders of magnitude faster than the second fastest algorithm, MAPA.

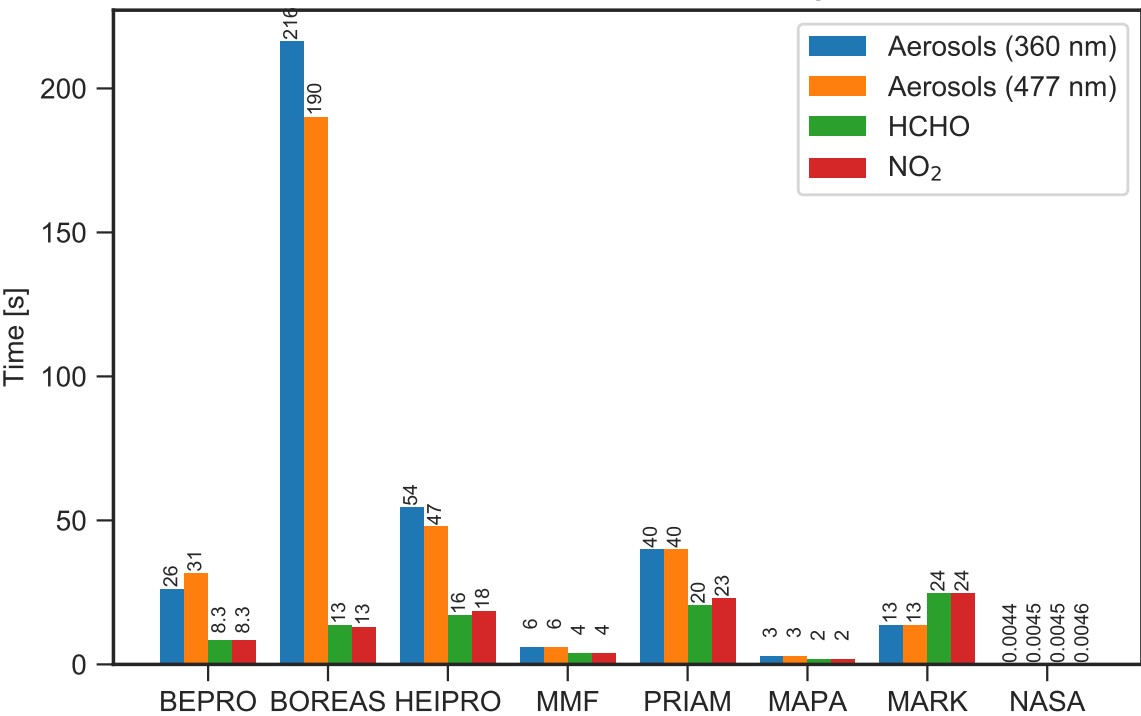

**Figure 22.** Time in seconds for the retrieval of a single profile on a single CPU core for each participant, separated by the target species as denoted in the legend.

## 7 Conclusions

Eight different algorithms for the retrieval of aerosol and trace gas vertical profiles from MAX-DOAS measurements have been compared under a large variety of atmospheric conditions by using synthetic measurements. Both OEM and parametrised algorithms, and also the analytic approach by NASA, show equally good performance in terms of the reproduction of the true atmospheric state, with a typical accuracy (in terms of RMS difference between true and retrieved state) of $(0.08 - 0.25)$ km$^{-1}$ for surface aerosol extinction, and $(5.9 - 15.0) \cdot 10^{10}$ molec/cm$^3$ (or about 2.4 - 6 ppb) for HCHO and NO$_2$ surface concentrations. These deviations, and also a potential positive bias towards the a priori for OEM algorithms, can be quite significant for clean air conditions, but are relatively small in polluted areas where several tens of ppb of NO$_2$ and HCHO, and aerosol extinction of up to 5 km$^{-1}$ can be present (e.g., Vlemmix et al., 2015b). Total columns of trace gases can be retrieved with a higher accuracy than for aerosols, with slopes $> 0.85$ and correlation coefficients $> 0.95$ in most cases. However, the accuracy is expected to be higher for real atmospheric measurements (e.g. Irie et al., 2011), since some of the scenarios within this study are quite arbitrary with the intention to test the algorithms under extreme conditions.

There are only few exceptions from this high level of agreement between the retrieved atmospheric state from the different algorithms. As a result of lack of convergence between true and modelled slant column densities, bePRO profiles are subject to a high degree of instability in the visible wavelength range for the aerosol scenarios AER0, AER8, AER9 and AER10 and to a lesser extent also AER4. Up to 25% of the bePRO profiles need to be discarded in order to achieve an accuracy similar to the other algorithms. However, bePRO performs well when convergence is reached. The synthetic data used for the study are not necessarily representative of real measurements, especially in terms of dSCDs errors, and sensitivity tests done by increasing the dSCDs errors but also previous publications (e.g. Hendrick et al., 2014; Vlemmix et al., 2015b) have shown that bePRO performs generally well with real measurement data, also in terms of convergence.

Aerosol AVKs from BOREAS differ from those of other OEM retrievals as additional regularisation is applied. They can therefore not be compared to those from the other retrievals and also do not comply with Equation 7. About 54% of the data from MAPA are flagged as invalid (39% of the moderate scenarios AER1 - AER7, and 93% of the cloud/fog scenarios AER8 - AER10), although the level of agreement of all MAPA profiles with the truth is comparable to the other algorithms, indicating that the MAPA flagging criteria might be too strict. However, the MAPA flagging successfully removes almost all outliers.

OEM algorithms tend to produce profiles biased towards the a priori, in particular at high altitudes where the sensitivity to the atmospheric state is small. OEM algorithms retrieving the logarithm of the target parameters show a higher degree of stability with less oscillations than those operating in linear space. Parametrised algorithms do not suffer from this disadvantage, but the possible results can be wide-spread when the sensitivity to the atmospheric state is low, which is particularly the case at high altitude or above layers with high extinction. However, despite these conceptual differences, the overall accuracy of OEM and parametrised algorithms is very similar.

Based on an analytical approach without using RTM calculations, the NASA algorithm is by far the fastest, with the retrieval of a single profile requiring less than $5\,\mu$s, followed by MAPA as the second fastest algorithm, requiring 2-3 s for a single retrieval based on LUTs. Involving on-line radiative transfer calculations, most OEM algorithms are slower, but their computational performance covers a wide range, mainly owing to the different approaches for the calculation of weighting functions. Being only about 2-3 times slower than the parametrised MAPA algorithm, MMF is by far the fastest OEM algorithm.

In summary, it can be concluded that, with only few exceptions, the algorithms presented here are capable of realistically retrieving aerosol and trace gas profiles in the lowermost $\approx 2$ km of the atmosphere, yielding 1.5 - 3.5 independent pieces of information depending on the target species and the atmospheric conditions. The comparison using synthetic measurements of course represents an idealisation in many respects, and the agreement between true and retrieved state might be worse for true atmospheric measurements. In particular, it has been assumed that forward model parameters, such as surface albedo, aerosol optical properties, etc., are perfectly well known, which is usually not the case for ambient measurements. Furthermore, the atmosphere has been assumed to be horizontally homogeneous, while in reality inhomogeneities, in particular broken clouds, can have a significant influence on MAX-DOAS measurements. Also, several instrumental aspects, such as a finite instrumental field of view, instrumental stray light, pointing inaccuracies or difficulties when pointing the telescope closely towards the Sun, were not considered here. Furthermore, the idealised set of atmospheric states considered in this synthetic intercomparison exercise does not cover all atmospheric conditions.

As a result of this study, the MMF and the MAPA algorithms, both showing best performance in terms of reconstruction of the atmospheric state and computational speed, were selected as profile algorithms for the FRM$_4$DOAS centralised near-real-time algorithm for a harmonised processing of MAX-DOAS data, which is planned to be made available to the community in the near future.

The median dSCDs compiled in this study, together with the corresponding atmospheric scenarios, are available online (http://frm4doas.aeronomie.be). They can serve as a reference dataset for future benchmarks of MAX-DOAS retrieval algorithms.

A detailed comparison of vertical profiles of trace gases and aerosols from MAX-DOAS field measurements performed during the CINDI-2 (http://www.tropomi.eu/data-products/cindi-2) campaign with co-located independent measurements, which comprises retrievals from 9 algorithms and 16 workgroups, will be the subject of a companion publication.

*Code availability.*    The bePRO and HEIPRO algorithms are available on request.

*Data availability.*    The reference database of synthetic dSCDs is available on the FRM$_4$DOAS webpage (http://frm4doas.aeronomie.be/index.php/documents).

*Acknowledgements.*    The funding of this study by the FRM$_4$DOAS project (ESA Contract No. 4000118181/16/I-EF) is gratefully acknowledged. The activities of the IUP Heidelberg were supported by the DFG project RAPSODI (grant No. PL 193/17-1). We are also thankful to
Marc Allaart (KNMI) for providing the ozonesonde data from De Bilt for constructing the temperature and pressure profiles used as input for the RT modelling.

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
