# Peer review of "Intercomparison of MAX-DOAS Vertical Profile Retrieval Algorithms: Studies using Synthetic Data."

_Atmospheric Measurement Techniques, 2018_

## Short Comment (SC1) · 2 Jan 2019

The studies conducted in this manuscript use synthetic MAX-DOAS measurement data as input to several different inversion algorithms in order to assess how well the assumed synthetic atmospheric state is reproduced by each individual inversion algorithm.

The underlying idea utilized here - using synthetic MAX-DOAS data to assess, and compare against, the output of an aerosol and trace gas inversion algorithm - seems strangely all too familiar to me (see Chapter 5: "MAX-DOAS inversion sensitivity studies", in: Zielcke, 2015). I am (slightly) surprised that my work is not mentioned here as a former colleague of several years of the lead author. Figure 1 of the manuscript also

appears to be "inspired" by my Figure 5.1 (p. 66, Zielcke, 2015).

While the saying goes "Imitation is the sincerest form of flattery", the following should be cited in the Introduction (1) or Methods (2) sections as at least one of the underlying foundations of this manuscript:

Zielcke, J. (2015): "Chapter 5: MAX-DOAS inversion sensitivity studies", in: Observations of reactive bromine, iodine and chlorine species in the Arctic and Antarctic with differential optical absorption spectroscopy, Dissertation, University of Heidelberg, http://doi.org/10.11588/heidok.00018932

Best regards, Johannes Zielcke

———————————————————

---

## Author Comment (AC1) · 21 Jan 2019

The thesis Johannes Zielcke (2016), of which our study is certainly not an imitation, shall be cited in a revised version of the manuscript among the numerous other publications that used synthetic measurements for the investigation of inverse models for aerosol and trace gas vertical profiles from MAX-DOAS measurements (e.g., Wagner et al., 2004; Frieß et al., 2006; Hay, 2010; Vlemmix et al., 2011; Yilmaz, 2012; Hartl and Wenig, 2013; Holla, 2013).

**References:**

Frieß, U., Monks, P., Remedios, J., Rozanov, A., Sinreich, R., Wagner, T., and Platt, U.: MAX-DOAS $O_4$ measurements: A new technique to derive information on atmospheric aerosols: 2. Modeling studies, J. Geophys. Res., 111, D14 203, doi: 10.1029/2005JD006618, 2006.

Hartl, A. and Wenig, M. O.: Regularisation model study for the least-squares retrieval of aerosol extinction time series from UV/VIS MAX-DOAS observations for a ground layer profile parameterisation, Atmos. Meas. Tech., 6, 1959-1980, doi: 10.5194/amt-6-1959-2013, 2013.

Hay, T.: MAX-DOAS measurements of bromine explosion events in McMurdo Sound, Antarctica, Ph.D. thesis, University of Canterbury, http://hdl.handle.net/10092/5394, 2010.

Holla, R.: Reactive Halogen Species above Salt Lakes and Salt Pans, Ph.D. thesis, University of Heidelberg, http://www.ub.uni-heidelberg.de/archiv/14636, 2013.

Vlemmix, T., Piters, A. J. M., Berkhout, A. J. C., Gast, L. F. L., Wang, P., and Levelt, P. F.: Ability of the MAX-DOAS method to derive profile information for $NO_2$: can the boundary layer and free troposphere be separated?, Atmos. Meas. Tech., 4, 2659-2684, doi:10.5194/amt-4-2659-2011, 2011.

Wagner, T., Dix, B., von Friedeburg, C., Frieß, U., Sanghavi, S., Sinreich, R., and Platt, U.: MAX-DOAS $O_4$ measurements: A new technique to derive information on atmospheric aerosols - principles and information content, J. Geophys. Res., 109, D22 205, doi: 10.1029/2004JD004904, 2004.

Yilmaz, S.: Retrieval of Atmospheric Aerosol and Trace Gas Vertical Profiles using Multi-Axis Differential Optical Absorption Spectroscopy, Ph.D. thesis, University of Heidelberg, http://www.ub.uni-heidelberg.de/archiv/13128, 2012.

Zielcke, J.: Observations of reactive bromine, iodine and chlorine species in the Arctic and Antarctic with Differential Optical Absorption Spectroscopy, Ph.D. thesis, University of Heidelberg, doi: 10.11588/heidok.00018932, https://archiv.ub.uni-heidelberg.de/volltextserver/18932/, 2015.

---

## Referee Comment (RC1) · Anonymous Referee #1 · 24 Jan 2019

**General comments**

The manuscript by Frieß et al. is well-written, well-structured and easy to follow. The authors provide a thorough intercomparison of eight different well-established or new MAX-DOAS vertical profile retrieval algorithms. The intercomparison is based on synthetic data generated from the medians of forward modelled SCDs of the participating radiative transfer models. The model outcomes for vertical aerosol and trace gas profiles are compared to the true states for a large set of typical distributions, covering a wide range of atmospheric conditions and also including extreme cases. The results of this study improve the assessment of the accuracy of MAX-DOAS vertical profile retrievals. Due to the extent of participating profile retrieval algorithms and the broad base of initial conditions the study will likely serve as a reference work for the MAX-

DOAS community. It will simplify the choice of appropriate retrieval algorithms in future projects and provides some clear recommendations, e.g. with respect to computational time or to use logarithmic retrievals for better stability.

Specific comments

Interesting findings are that horizontally homogeneous clouds in the free troposphere have little impact on the sensitivity of MAX-DOAS retrievals, the increased information content for trace gases caused by uplifted aerosol layers around 1 km and the weak dependency of the stability of the trace gas retrievals on the exact extinction profile. It is valuable to see that averaging kernels are quite similar for different positions of the sun.

p. 22, l. 2: Which sensitivity studies, reference?

Technical corrections

p.29, fig 16, l. 3: ... as blue vertical line -> ... as blue horizontal line p.31, l. 4: discrepancies are mainly occur -> discrepancies mainly occur p.36, l. 4: between retrieved and atmospheric state p.36, l. 10: ... have shown ... p.36, l. 13: About 54% of the data ...

---

## Referee Comment (RC2) · Anonymous Referee #2 · 9 Feb 2019

General Comments

This manuscript presents a detailed intercomparison of the performance of eight algorithms used to retrieve vertical profiles of aerosol, HCHO, and NO2 in the atmospheric boundary layer from MAX-DOAS slant column densities. A set of synthetic measurements is first generated from the medians of seven simulated datasets of slant column densities simulated using five radiative transfer models. Simulations are performed for 11 aerosol profiles, 10 trace gas profiles, and multiple viewing geometries (11 elevation angles, three solar zenith angles, three relative azimuth angles), resulting in 990 O4 SCDs at 360 nm and at 477 nm, and 9900 SCDs each for HCHO and NO2.

The simulated SCDs from the different RTMs agree well under all conditions. The retrievals use three approaches: five algorithms are based on the optimal estimation

method, two on parametrised methods, and one on an analytical method. Differences between the retrieved vertical profiles, surface concentrations, and total columns obtained from the eight algorithms are quantified and generally agree, with the exception of some outliers. The retrievals result in DOFS of 1.5-3.5 in the lowest 2 km of the atmosphere, depending on the target species and the atmospheric conditions. Limitations resulting from the use of synthetic measurements are identified, and so these results may be interpreted as a best-case scenario relative to retrievals using atmospheric measurements with real instruments.

This study is part of the Fiducial Reference Measurements for Ground-Based DOAS Air-Quality Observations (FRM4DOAS) project, and the results provide a benchmark reference for the development of a community algorithm for a harmonised near-real-time processing of MAX-DOAS data.

The manuscript is straightforward and well-written, providing a clear and concise description of the intercomparison strategy, the algorithms, the RTM simulations, and the results of the retrieval intercomparisons. I have only minor comments, and I recommend publication after these are addressed.

Specific Comments

Page 1, line 12 – Here, or in the main text (e.g., pages 35-36), provide some context for these RMS differences. Are these numbers negligible or significant? How do they compare to typical absolute values?

The multi-panel figures include a lot of information, but it is difficult to read some of the text (e.g., the numbers in the panels of Figures 5-7) and to distinguish some of the features discussed in the text. I suggest looking at all of the multi-panel figures and enlarging text or making other revisions where possible to make them clearer and more easily readable.

Technical Corrections

Page 1, line 11 – root mean square (not squares)

Page 1, line 12 – between THE true . . .

Page 1, line 16 – emissions. Monitoring . . .

Page 2, line 19 – here and elsewhere, hyphenate "state-of-the-art"

Page 2, line 23 – straightforward

Page 3, line 7, 24, 24, etc. – Here and for all equations, include punctuation – add periods to equations where relevant.

Page 3, line 15 – define variables s and rho in Eqn 3

Page 4, line 8 – DOFS is more commonly used than DFS for Degrees Of Freedom for Signal

Page 4, line 20 – well-established

Page 4, line 22 – should AVL be AVK?

Page 5, Figure 1 – Should rightmost green box say "Reference dataset for dSCDs" rather than SCDs? Line 12 says that the dSCDs are the reference dataset.

Page 5, line 15 – reorder as: This dataset is referred to below as the . . .

Page 6, Table 1 – All acronyms should be defined here or in the text.

Page 7, line 22 and page 10, line 10 – a-priori is hyphenated here but nowhere else

Page 7, line 29 – grid points, (add comma) . . . layer, which IS considered . . .

Page 8, line 23 – define MMF (it's not defined anywhere – also check that all acronyms are defined throughout)

Page 9, line 16 – and second, (add comma)

Page 9, line 18,19 – move definition of (RAA) immediately after "relative azimuth angles"

Page 9, line 28,29 – allows the uncertainty of the resulting profiles to be determined.

Page 10, line 3 – quantify what "level of agreement" is used in flagging data

Page 10, line 12 – (2011, 2015a)

Page 10, line 19 – a-posteriori is hyphenated here but nowhere else

Page 10, line 26 – algorithm WAS developed . . .

Page 11, line 14 – atmospheric radiative transfer models

Page 11, line 19 – for the forward MODELLING than . . .

Page 11, line 31 – during THE CINDI-2 . . .

Page 12, Figure 2 – Add a reference in the figure caption to Tables 2 and 3 for definitions of the profiles indicated in the legends.

Page 14, line 1 – Table 5. RTM parameterS for . . .

Page 14, line 7 – serve as A reference

Page 14, line 13 – as THE forward

Page 14, line 14 – in THE case

Page 14, line 19 – "some of" or "a portion of" rather than "parts of" ?

Page 14, line 20 – representationS

Page 15, Figure 3 – Add more tick marks to the horizontal axes.

Page 16, Figure 4 caption – root mean square (no S) ... correlation BETWEEN the . . . compared to the medianS from . . .

Page 17, line 9 – similar magnitude to the

Page 17, line 14 – trace gas profileS

Page 17, line 15 – azimuth angleS

Page 17, line 22 – interpolated onTO the

Page 19,20 Figure 7 and 8 captions – Same as Figure 5 . . .

Page 21, line 5 – RMS has already been defined – delete "root mean squares difference"

Page 21, line 7 – problems retrievING the

Page 23, line 2 – in THE case of

Page 27, line 4,5 – requires, however, excludING

Page 28, line 12 – values FOR altitudes

Page 28, line 30 – as is the case (delete "it")

Page 28, line 31 – prevents THE retrievAL of negative

Page 28, line 33 – in THE case of

Page 29, Figure 16 – The label on the top of the left column says "AOD" but the caption refers to "AOT". Choose and use one consistently throughout.

Page 29, Figure 16, last line of caption – all data ARE shown

Page 30, Figure 17 caption – not correct to say this is the same as Figure 16. This uses the TG profiles, while Figure 16 uses the AER profiles. Revise.

Page 30, line 2 – high-altitude cloud

Page 32, Figure 19, last line of caption – all data ARE shown

Page 35, line 6 – algorithm, MAPA. (add comma)

Page 36, line 10 – have showN that bePRO generally performs well with . . .

Page 36, line 20 – which is particularLY the case

Page 37, line 4 – median dSCDs (add s) . . . ARE available

Page 37, line 5 – THEY can serve as

Page 37, line 7 – delete definition of CINDI-2 as it was defined on page 10

Page 37, line 10 – available on REQUEST. (what about the availability of the other algorithms?)

---

## Referee Comment (RC3) · Anonymous Referee #3 · 16 Feb 2019

This manuscript describes intercomparison of MAX-DOAS vertical profile retrieval algorithms for O4 (aerosols), NO2, and HCHO, based on synthetic SCD data. The algorithms include OEM, parameterization, and analytical approaches. Ensemble means of SCD were synthesized using forward RTM calculations with prescribed aerosol/gas profiles, instead of using measurements, and then they were inverted by the algorithms to yield vertical profiles, which were finally compared with the given original profiles. From large number of experiments where various aerosol and gas concentration profiles were assumed, deviations were systematically analyzed. Causes of deviations were sometimes identified. The methodology is new, forming a closure, and sounds robust. The results include important implication and the new knowledge obtained will be reflected into key revision of algorithms in the future. However, at certain times I

found points needing clarification. For example, I wonder how the NASA algorithm using only RTM calculations with Rayleigh scattering, works to yield aerosol profiles; O4 optical depth could be assigned for each layer, but I believe only RTM calculations with aerosols can connect the O4 information and aerosols. Secondly degrees of agreement with surface values (section 6.6) should be compared to those with vertical total columns. The last sentence of Abstract and the second sentence of Conclusions focused on difference quantification of "concentrations", but implication for column values would be similarly or even more important, considering MAX-DOAS is heavily used for satellite validation. Clarification on several other points listed below should also be made. Overall, I recommend publication after minor revision.

Specific points:

1. Page 1, line 12. The authors state that the values are root mean squares, but they are not described much in detail in text (section 6.6).

2. Page 2, line 34. A posteriori modelled "d"SCDs? Same for the rightmost green box in Figure 1?

3. Page 3, equation (4): What is $S_e$? How were they assumed in the OEM calculations? The dSCD errors listed in Table 6 correspond to this?

4. Page 10-11. Description of NASA algorithm should be elaborated as mentioned earlier. In Page 11, line 2, Are there cases where less than four measurements are available, for this synthetic-data-based study? Page 11, line 6. O4 dSCDs at low angles are used instead of the aerosol retrieval results – how this approach works without RTMs is difficult to understand.

5. AER8, 9, and 10 in Table 3 and 5. Parameters for fog/clouds (particularly for single scattering albedo) are same as those for aerosols?

6. Bottom panels of Figure 3 for gases: Results for all scenarios with different aerosol profiles are included?

7. Page 20, lines 1-3. The authors should mention that O4 profile shape is heavily weighted to low layers.

8. Page 20, line 13. Better specifically mention as "synthetic" measurement vector (y)? Same for x axis label of Figure 9. Y-axis label in Figure 9 should be better mentioned as a posteriori modeled dSCDs?

9. Page 20, line 3. Any reasons for the underestimation of O4 at 477 nm?

10. Figure 10 and 15. Y-Axis range for Slope is better zoomed to narrower range?

11. Figure 13 caption. Legend for coloured symbols corresponding to aerosol scenario is not found.

12. Figure 14. My guess is that MAX-DOAS gas determination would be difficult in the AER2+TG7 scenario. Can the scenario be identified in the plot for some discussion?

13. Page 28, lines 1-5. How the failure in retrievals in aerosol/cloud affected the gas retrievals? If correct aerosols profiles are given, better agreement is obtained?

14. AER0 and TG0 cases for bePRO, HEIPRO, MMF, and PROAMF in Figures 16 and 17. As mentioned in page 30 lines 7-8, a positive bias is present. This might be fatal for satellite validation in clean region. Can they be easily screened out during post error analysis, for example, comparison with largest dSCD values?

15. Y-axis of Figure 16. AOT is used throughout text?

16. Page 33. Degrees of agreement with surface values (section 6.6) should be compared to those with vertical total columns.

17. Section 6.7. Any time loss during data I/O for some algorithms?

18. Page 36, line 7. bePRO

19. Page 36, line 10. have shown

20. Page 37, line1. Do the authors mean "direct-sun" observation here?

[Figure]

21. Recommendation learnt from this study should be listed in Conclusions?

---

## Author Comment (AC2) · 22 Mar 2019

We thank Referee #1 for his positive comments on our manuscript. We reply to the individual comments point by point, with the original comments shown in *italic*, our replies in roman, and changes to the manuscript in **bold**.

*p. 22, l. 2: Which sensitivity studies, reference?*

These were internal unpublished studies performed at BIRA, during which the stability of the profile retrieval has been tested using either a Gauss-Newton or a Levenberg-Marquardt inversion scheme. They were motivated by the lack of convergence for some of the scenarios within this synthetic study. We feel that it is beyond the scope of this paper to investigate the internal details of the implementations of the individual

algorithms and would therefore rather not elaborate this in more detail. a modification of the inversion scheme within bePRO is not foreseen since bePRO will be replaced by the new MMF algorithm in the future.

*p.29, fig 16, l. 3: ... as blue vertical line → ... as blue horizontal line*

Done

*p.31, l. 4: discrepancies are mainly occur → discrepancies mainly occur*

Done

*p.36, l. 4: between retrieved and atmospheric state*

This has been replaced by **between the retrieved atmospheric state from the different algorithms**.

*p.36, l. 10: ... have shown ...*

*p.36, l. 13: About 54% of the data ...*

Done

---

## Author Comment (AC3) · 22 Mar 2019

We thank Referee #2 for his thoughtful and positive comments, as well as the careful proofreading. We reply to the individual comments point by point, with the original comments shown in *italic*, our replies in roman, and changes to the manuscript in **bold**.

*Page 1, line 12 - Here, or in the main text (e.g., pages 35-36), provide some context for these RMS differences. Are these numbers negligible or significant? How do they compare to typical absolute values?*

It is indeed useful to bring the deviations into relation with typical atmospheric values - although these can be very variable. The respective statement in the conclusions

has been replaced with: **Both OEM and parametrised algorithms, and also the analytic approach by NASA, show equally good performance in terms of the reproduction of the true atmospheric state, with a typical accuracy (in terms of RMS difference between true and retrieved state) of** $(0.08 - 0.25)$ **km$^{-1}$ for surface aerosol extinction, and** $(5.9 - 15.0) \cdot 10^{10}$ **molec/cm$^3$ (or about 2.4 - 6 ppb) for HCHO and NO$_2$ surface concentrations. These deviations, and also a potential positive bias towards the a priori for OEM algorithms, can be quite significant for clean air conditions, but are relatively small in polluted areas where several tens of ppb of NO$_2$ and HCHO, and aerosol extinction of up to 5 km$^{-1}$ can be present (e.g., Vlemmix et al., 2015). Total columns of trace gases can be retrieved with a higher accuracy than for aerosols, with slopes** $> 0.85$ **and correlation coefficients** $> 0.95$ **in most cases. However, the accuracy is expected to be higher for real atmospheric measurements (e.g. ?), since some of the scenarios within this study are quite arbitrary with the intention to test the algorithms under extreme conditions.**

*The multi-panel figures include a lot of information, but it is difficult to read some of the text (e.g., the numbers in the panels of Figures 5-7) and to distinguish some of the features discussed in the text. I suggest looking at all of the multi-panel figures and enlarging text or making other revisions where possible to make them clearer and more easily readable.*

Figures 5-8, and 11-14 indeed contain a lot of information, but we are convinced that it is important to show the averaging kernels and vertical profiles for each of the aerosol and trace gas scenarios in order to give the reader an impression of the performance of the algorithms under different atmospheric conditions. In order to improve the readability, we have made the following modifications to the figures: (1) Increase of font size for the degrees of freedom for signal in Figures 5-8; (2) Increase of font size for the RMS in Figures 11-14; (3) Larger legends in Figures 11-14, now moved to the top of the figures.

*Page 1, line 11 - root mean square (not squares)*

Done (here and anywhere else)

*Page 1, line 12 - between THE true...*

Done

*Page 1, line 16 - emissions. Monitoring*

Done

*Page 2, line 19 - here and elsewhere, hyphenate "state-of-the-art"*

Done

*Page 2, line 23 - straightforward*

Done

*Page 3, line 7, 24, 24, etc. - Here and for all equations, include punctuation - add periods to equations where relevant.*

Our impression is that it is rather unusual to add a period at the end of a single-line equation even if the sentence ends there, as this would cause an ambiguity as that the period might be interpreted as part of the equation.

*Page 3, line 15 - define variables s and rho in Eqn 3*

The following sentence has been added: **Here, $\rho$ is the number density of the trace gas and $s$ parametrises the light path length through the atmosphere.**

*Page 4, line 8 - DOFS is more commonly used than DFS for Degrees Of Freedom for Signal*

To our knowledge, both DOFS and DFS are used in the literature as abbreviation for the degrees of freedom for signal. For the sake of consistency, we would rather prefer to use DFS since this is the term that has been used by the main author in the past

(Frieß et al., 2006, 2011, 2016).

*Page 4, line 20 - well-established*

Done

*Page 4, line 22 - should AVL be AVK?*

Yes, this has been corrected

*Page 5, Figure 1 - Should rightmost green box say "Reference dataset for dSCDs" rather than SCDs? Line 12 says that the dSCDs are the reference dataset.*

This is correct, **SCDs** has been replaced with **dSCDs** in Figure 1.

*Page 5, line 15 - reorder as: This dataset is referred to below as the ...*

Done

*Page 6, Table 1 - All acronyms should be defined here or in the text.*

All acronyms for institutes, retrieval algorithms and RTMs are now defined in the description of the algorithms (Sections 3.1 - 3.8)

*Page 7, line 22 and page 10, line 10 - a-priori is hyphenated here but nowhere else*

The hyphen has been removed

*Page 7, line 29 - grid points, (add comma) ... layer, which IS considered ...*

Done

*Page 8, line 23 - define MMF (it's not defined anywhere - also check that all acronyms are defined throughout)*

See the reply to your comment above. In addition, a reference to the recently published AMTD paper describing MMF algorithm has been added.

*Page 9, line 16 - and second, (add comma)*

Done

*Page 9, line 18,19 - move definition of (RAA) immediately after "relative azimuth angles"*

Done

*Page 9, line 28,29 - allows the uncertainty of the resulting profiles to be determined.*

Done

*Page 10, line 3 - quantify what "level of agreement" is used in flagging data*

The agreement between forward modeled and measured dSCDs is judged based on the dSCD error. Details can be found in Beirle et al. (2018). The description of the MAPA algorithm has been modified as follows: **Flagging is based on different criteria: (1) The level of agreement between forward model and measurement as compared to the dSCD error (for details see Beirle et al., 2018), ...**

*Page 10, line 12 - (2011, 2015a)*

Done

*Page 10, line 19 - a-posteriori is hyphenated here but nowhere else*

The hyphen has been removed

*Page 10, line 26 - algorithm WAS developed ...*

Done

*Page 11, line 14 - atmospheric radiative transfer models*

Done

*Page 11, line 19 - for the forward MODELLING than ...*

Done

*Page 11, line 31 - during THE CINDI-2 ...*

Done

*Page 12, Figure 2 - Add a reference in the figure caption to Tables 2 and 3 for definitions of the profiles indicated in the legends.*

The following sentence has been added to the caption of Figure 2: **The properties of the individual profiles are described in Tables 2 and 3.**

*Page 14, line 1 - Table 5. RTM parameterS for ...*

Done

*Page 14, line 7 - serve as A reference*

Done

*Page 14, line 13 - as THE forward*

Done

*Page 14, line 14 - in THE case*

Done

*Page 14, line 19 - "some of" or "a portion of" rather than "parts of" ?*

Replaced with **some of**

*Page 14, line 20 - representationS*

Done

*Page 15, Figure 3 - Add more tick marks to the horizontal axes.*

The number of x-axis ticks has been increased in Figure 3

*Page 16, Figure 4 caption - root mean square (no S) ... correlation BETWEEN the ...*

*compared to the medianS from ...*

This sentence has been replaced with **Slope, intercept, regression coefficient and root mean square difference (RMS) of the correlation between the SCDs from the individual forward models and the median SCDs from all models.**

*Page 17, line 9 - similar magnitude to the*

We feel that the preposition "as" is more appropriate here.

*Page 17, line 14 - trace gas profileS*

Done

*Page 17, line 15 - azimuth angleS*

Done

*Page 17, line 22 - interpolated onTO the*

Done

*Page 19,20 Figure 7 and 8 captions - Same as Figure 5 ...*

Done

*Page 21, line 5 - RMS has already been defined - delete "root mean squares difference"*

Done

*Page 21, line 7 - problems retrievING the*

Done

*Page 23, line 2 - in THE case of*

Done

*Page 27, line 4,5 - requires, however, excludING*

Done

*Page 28, line 12 - values FOR altitudes*

Done

*Page 28, line 30 - as is the case (delete "it")*

Done

*Page 28, line 31 - prevents THE retrievAL of negative*

Done

*Page 28, line 33 - in THE case of*

Done

*Page 29, Figure 16 - The label on the top of the left column says "AOD" but the caption refers to "AOT". Choose and use one consistently throughout.*

We intend to use AOT throughout the paper since the term 'density' usually refers to an intensive variable, but AOT is an extensive quantity. **AOD** has been replaced with **AOT** in Figure 16.

*Page 29, Figure 16, last line of caption - all data ARE shown*

Done

*Page 30, Figure 17 caption - not correct to say this is the same as Figure 16. This uses the TG profiles, while Figure 16 uses the AER profiles. Revise.*

The legend has been replaced with **Box-whisker plots of the retrieved HCHO (left) and NO$_2$ (right) VCD for the different trace gas scenarios (rows) and retrieval algorithms (columns) based on retrievals using noisy dSCDs. Crosses show the mean, dashed horizontal lines the median. Shaded areas indicate the (25% -75%) percentile, whiskers the (5% - 95%) percentile. The true VCD is shown as blue**

**vertical line. For each algorithm, all data is shown on the left in black, and data marked as valid on the right in green. Also shown is the percentage of valid data. The median for each aerosol scenario is shown as coloured symbol as indicated in the legend.**

*Page 30, line 2 - high-altitude cloud*

Done

*Page 32, Figure 19, last line of caption - all data ARE shown*

Done

*Page 35, line 6 - algorithm, MAPA. (add comma)*

Done

*Page 36, line 10 - have showN that bePRO generally performs well with ...*

Done

*Page 36, line 20 - which is particularLY the case*

Done

*Page 37, line 4 - median dSCDs (add s) . . . ARE available*

Done

*Page 37, line 5 - THEY can serve as*

Done

*Page 37, line 7 - delete definition of CINDI-2 as it was defined on page 10*

Done

Page 37, line 10 - available on REQUEST. (what about the availability of the other algorithms?)

This has been corrected. The other algorithms are not (yet) readily available.

---

## Author Comment (AC4) · 22 Mar 2019

We thank Referee #3 for his detailed comments. We reply to the individual comments point by point, with the original comments shown in *italic*, our replies in roman, and changes to the manuscript in **bold**.

*However, at certain times I found points needing clarification. For example, I wonder how the NASA algorithm using only RTM calculations with Rayleigh scattering, works to yield aerosol profiles; O4 optical depth could be assigned for each layer, but I believe only RTM calculations with aerosols can connect the O4 information and aerosols.*

The description of the NASA algorithm has been updated with a more detailed explanation of the approaches for aerosol and trace gase profile retrieval. The revised

[Figure]

NASA algorithm description has been uploaded to the public discussion page as a supplemental document.

*Secondly degrees of agreement with surface values (section 6.6) should be compared to those with vertical total columns. The last sentence of Abstract and the second sentence of Conclusions focused on difference quantification of "concentrations", but implication for column values would be similarly or even more important, considering MAX-DOAS is heavily used for satellite validation.*

In order to bring the accuracy of surface value and total column into relation, we have added the following statement to Section 6.6: **For the prescribed profile scenarios, retrieved aerosol surface extinction (mean regression coefficient from all algorithms $\bar{R} = 0.903$ at 360 nm and $\bar{R} = 0.825$ at 477 nm ) shows a better agreement with the true value that the AOT ($\bar{R} = 0.874$ at 360 nm and $\bar{R} = 0.803$ at 477 nm). The opposite is true for trace gases, where the agreement of the total column ($\bar{R} = 0.969$ and $\bar{R} = 0.955$ for HCHO and NO$_2$, respectively) is better than for the surface mixing ratio ($\bar{R} = 0.894$ and $\bar{R} = 0.780$).**

*1. Page 1, line 12. The authors state that the values are root mean squares, but they are not described much in detail in text (section 6.6).*

We have added a paragraph discussing the RMS of surface and total column values to Section 6.6, see above.

*2. Page 2, line 34. A posteriori modelled "d"SCDs? Same for the rightmost green box in Figure 1?*

**SCDs** has been replaced with **dSCDs**, both in the text and in Figure 1.

*3. Page 3, equation (4): What is $S_e$? How were they assumed in the OEM calculations? The dSCD errors listed in Table 6 correspond to this?*

We have added the following statement after Figure 4: **Here $S_\epsilon$ is the measurement covariance matrix, which, under the assumption that the measurements are in-**

**dependent, is a matrix with the squares of the measurement errors (specified in Section 6.1 and Table 6) as diagonal elements and zero values elsewhere.**

*4. Page 10-11. Description of NASA algorithm should be elaborated as mentioned earlier. In Page 11, line 2, Are there cases where less than four measurements are available, for this synthetic-data-based study? Page 11, line 6. O4 dSCDs at low angles are used instead of the aerosol retrieval results ? how this approach works without RTMs is difficult to understand.*

The NASA algorithm description has been updated, see our reply to the general comment above.

*5. AER8, 9, and 10 in Table 3 and 5. Parameters for fog/clouds (particularly for single scattering albedo) are same as those for aerosols?*

Yes, the optical parameters are the same. We are aware that in reality optical properties of cloud droplets are significantly different from those of aerosols, and there is of course also a high variability for different kinds of aerosols. However, it is assumed here that aerosol properties are perfectly known and are the same for forward modelling and inversion. So the choice of the optical properties should have very little influence on the results.

*6. Bottom panels of Figure 3 for gases: Results for all scenarios with different aerosol profiles are included?*

Yes, the following sentence has been added to the caption of Figure 3: **For HCHO and NO$_2$, results for all scenarios with different aerosol profiles are included.**

*7. Page 20, lines 1-3. The authors should mention that O4 profile shape is heavily weighted to low layers.*

We have added the following sentence to the discussion of the vertical sensitivity in the first paragraph of Section 6.2: **This is a result of the measurement geometry and, in case of aerosols, also of the fact that the O$_4$ vertical distribution is heavily**

**weighted to the surface.**

*8. Page 20, line 13. Better specifically mention as "synthetic" measurement vector (y)?
Same for x axis label of Figure 9. Y-axis label in Figure 9 should be better mentioned
as a posteriori modeled dSCDs?*

The sentence on page 20, line 13, stating that the agreement between the measurement vector $\mathbf{y}$ and the measurement vector $\mathbf{F}(\hat{\mathbf{x}})$ is an important indicator for the level of convergence, is valid not only for synthetic studies but in general for all applications. We would therefore rather not add the term "synthetic". It is stated in the caption of Figure 9 that the y-axis shows a posteriori dSCDs, and adding this to the y-axis label would make the label too lengthy. To make clear that we refer to simulated measurements from the synthetic dataset whenever we state "measurements", we have added the following statement to Section 6.1: **In the following, we refer to this synthetic dataset as the measured dSCDs.**

*9. Page 20, line 3. Any reasons for the underestimation of O4 at 477 nm?*

We assume that you refer to page 21, line 3. One of the reasons might be that the constraint of the retrievals by the a priori covariance might be too strong specifically for the PriAM algorithm. However, the investigation of the reasons for deficiencies of the individual algorithms is beyond the scope of this study, and we have no knowledge about the reasons for the underestimation of $O_4$ at 477 nm by the PRIAM algorithm.

*10. Figure 10 and 15. Y-Axis range for Slope is better zoomed to narrower range?*

Our intention was to have the same y-axis ranges for slope and regression coefficients in Figures 4, 10, 15, 18, and 21. This has now been changed and the axis ranges now better match the range of the data points.

*11. Figure 13 caption. Legend for coloured symbols corresponding to aerosol scenario
is not found.*

The legend now has moved to the top of the Figure, with a larger font size for better

readability.

*12. Figure 14. My guess is that MAX-DOAS gas determination would be difficult in the AER2+TG7 scenario. Can the scenario be identified in the plot for some discussion?*

The AER2/TG7 scenarios (shown as green circles) do not pose any particular difficulty for any of the algorithms. In contrast, some of the OEM algorithms (BEPRO and BOREAS) have difficulties with uplifted profiles during fog (AER8/TG7; green diamonds).

*13. Page 28, lines 1-5. How the failure in retrievals in aerosol/cloud affected the gas retrievals? If correct aerosols profiles are given, better agreement is obtained?*

In general, the trace gas retrieval is surprisingly stable even if the aerosol profiles appear to be unrealistic, a finding that has also been confirmed by real measurements. We focused on the performance of the algorithms based on the processing chain that is also applied to field measurements (i.e., using the aerosol profiles from the retrieval based on $O_4$ dSCDs as a constraint for the trace gas retrieval), and tests using the 'true' aerosol profile as a constraint for the trace gas retrieval are beyond the scope of this study.

*14. AER0 and TG0 cases for bePRO, HEIPRO, MMF, and PROAMF in Figures 16 and 17. As mentioned in page 30 lines 7-8, a positive bias is present. This might be fatal for satellite validation in clean region. Can they be easily screened out during post error analysis, for example, comparison with largest dSCD values?*

The bias for low (or zero) aerosol and trace gas scenarios from the OEM algorithms is indeed quite large for the settings chosen in this study. For atmospheric measurements, this can be overcome by either using smaller a priori concentrations/extinctions during clean air conditions and/or by increasing the a priori covariance. It is, however, crucial to investigate potential biases and other error sources, and to minimise these by choosing optimal settings for each individual field site and for each algorithm.

[Figure]

*15. Y-axis of Figure 16. AOT is used throughout text?*

AOD has been replaced by AOT in Figure 16.

*16. Page 33. Degrees of agreement with surface values (section 6.6) should be compared to those with vertical total columns.*

See our reply to the general comment above.

*17. Section 6.7. Any time loss during data I/O for some algorithms?*

Yes, the time for data I/O has been included in the benchmark. This should, however, only be significant for the NASA algorithm with a processing time in the order of milliseconds.

*18. Page 36, line 7. bePRO*

This has been corrected.

*19. Page 36, line 10. have shown*

This has been corrected.

*20. Page 37, line1. Do the authors mean "direct-sun" observation here?*

We are not referring to direct sun observations here, but to observations close to the sun (e.g., in the aureole region), which can cause difficulties due to stray light within the telescope, or unintentional intersection of the field of view with the solar disc. Also, the aureole region is particular sensitive to the aerosol phase function and a wrong choice of the aerosol optical parameters for the retrieval might introduce larger errors than for measurements elsewhere in the sky.

*21. Recommendation learnt from this study should be listed in Conclusions?*

The following statement has been added to the conclusions: **As a result of this study, the MMF and the MAPA algorithms, both showing best performance in terms of reconstruction of the atmospheric state and computational speed,**

[Figure]

**were selected as profile algorithms for the FRM$_4$DOAS centralised near-real-time algorithm for a harmonised processing of MAX-DOAS data, which is planned to be made available to the community in the near future.**

Please also note the supplement to this comment:
https://www.atmos-meas-tech-discuss.net/amt-2018-423/amt-2018-423-AC4-supplement.pdf

**Supplement:**

**3.8 The NASA algorithm**

The National Aeronautics and Space Administration (NASA) Real Time algorithm is developed as a quick look algorithm that relies on the fact that atmospheric scattering strongly affects DOAS measured $O_4$ absorption (Spinei et al., Fast aerosol extinction coefficient profile estimation from MAXDOAS UV/VIS measurements, in preparation). Two separate approaches are used for aerosol and trace gas profile retrieval. The aerosol profile algorithm determines the layer aerosol extinction coefficients by comparing measured Ring and $O_4$ absorption with Ring and $O_4$ absorption under pure Rayleigh conditions. Air mass factors and Ring absorption for the Rayleigh case are pre-calculated using the VLIDORT v2.8 and LIDORT-LRRS v2.5 radiative transfer models, respectively (Spurr et al., 2008; Spurr, 2008) assuming the U.S. standard atmosphere. Since Ring simulations were not provided in this study, aerosol analysis was performed only at 477 nm. $O_4$ dSCDs are corrected for SZA dependence. Eq. 10 is the simplified equation used in this study to calculate aerosol scattering extinction coefficients at each layer for specific observation geometry (EA and RAA) $\Theta$. We also assume an aerosol single scattering albedo of $\omega_{aer}(\lambda) = 1$.

$$\epsilon_{aer}(\lambda, \Theta, \vartheta) \approx \frac{\tau_{O4}^{\mathrm{noaer}} - \tau_{O4}^{\mathrm{aer}}}{\Delta h} \tag{10}$$

with $\tau_{O4}^{\mathrm{aer}}$ and $\tau_{O4}^{\mathrm{noaer}}$ being the optical density with and without aerosols in the respective layer, $\lambda$ denoting wavelength, and $\vartheta$ the SZA. The thickness $\Delta h$ of the respective layer is determined from the corrected $O_4$ dSCD using simple trigonometry according to Eq. 11 and 12, resulting in an atmosphere specific grid:

[revised manuscript text omitted]